# Addressing Exogenous Variability in Cooperative Multi-Agent Reinforcement Learning

## Abstract

Multi-agent reinforcement learning (MARL) has advanced control of many cooperative multi-agent systems. However, most approaches are trained against a single fixed adversarial strategy, leaving teams fragile to adversarial strategy shifts at test time. To handle such limitations, in this paper, we recast cooperative MARL from a new perspective into an *Exogenous Dec-POMDP*, separating agent-controllable endogenous and environment-driven exogenous dynamics in order to learn policies that adapt to exogenous shifts while preserving coordination. Our framework is composed of two main components: (i) learning exogenous dynamics and (ii) updating policy with two complementary goals - coordination to achieve high team return and causal influence on future exogenous evolution. We implement the framework under centralized training with decentralized execution into a practical algorithm, named Learning Exogenous Influence for Coordination and Adaptation (**LEICA**), and evaluate it on SMAX with distinct train/test adversarial strategies. Experimental results show that our approach drastically improves performance in test time with unseen opponents' strategies while achieving high training-time performance, demonstrating its ability to handle exogenous shift and improve training stability.

## 1 Introduction

Multi-Agent Reinforcement Learning (MARL) has achieved impressive success across a wide range of domains including games (Silver et al., 2016; Vinyals et al., 2019), autonomous robotics (Pinto et al., 2017; Gupta et al., 2017), and large-scale decision-making (Lowe et al., 2017; Wang et al., 2020). In particular, cooperative benchmarks such as the StarCraft Multi-Agent Challenge (SMAC) (Samvelyan et al., 2019) designed to model battles against opponent teams, have spurred the development of various MARL algorithms including value decomposition methods (Sunehag et al., 2017; Rashid et al., 2018) and actor-critic methods (Yu et al., 2022; Zhong et al., 2024). These algorithms demonstrate that coordinated multi-agent behaviors can be learned efficiently under controlled and well-defined environments.

Despite such progresses, current cooperative MARL methods remain brittle under exogenous variations, such as changes in opponent strategies, scenario parameters or latent environment regimes, which are not controlled by the team but alter dynamics (Mahajan et al., 2022). Typically, policies trained on a narrow distribution frequently overfit coordination to training regimes and degrade when the exogenous factors shift at test time (Deng et al., 2024); the restricted diversity of opponent dynamics often leads algorithms to exploit static weaknesses rather than acquire robust coordination skills (Mitra, 2024). Due to the inherent non-stationarity of multi-agent settings induced by concurrently-acting agents (Papoudakis et al., 2019), the disentanglement of endogenous and exogenous sources of variation is challenging with conventional MARL formulations. This necessitates a new MARL formulation and learning rule that explicitly separates controllable and uncontrollable factors in dynamics and synergizes inter-agent coordination and adaptation to exogeous variations.

To this end, we newly define an *Exogenous Dec-POMDP (ED-POMDP)*, which views the global state as being partitioned into team-controllable endogenous components and environment/opponent-driven exogenous components. The new formulation has two advantages. First, it enables more effective inference of exogenous dynamics by explicitly separating exogenous influences. Second, it can guide policy learning with two complementary goals: (i) a coordina-

tion goal that prioritizes endogenous changes according to their impact on return and (ii) an influence goal that promotes policy updates according to the causal influence of endogenous changes on subsequent exogenous evolution. These goals jointly drive policies to both sustain efficient team coordination and adapt to shifting exogenous regimes. Under centralized training and decentralized execution (CTDE), we propose a learning scheme to realize the aforementioned goals under ED-POMDP.

Our contributions are summarize as follows:

- We formalize the ED-POMDP under a broad distribution of environmental variations.

- We propose a learning framework for this new formulation with sensitivity-based influence weighting with counterfactual endogenous change.

- We build a new scalable SMAX (Rutherford et al., 2024) evaluation benchmark with controlled exogenous variation across multiple tasks and train-test regimes for ED-POMDP, and demonstrate consistent generalization gains of the proposed method over existing baselines with this benchmark.

## 2 BACKGROUND

A cooperative multi-agent environment is formulated as a Decentralized Partially-Observable Markov Decision Process (Dec-POMDP) (Bernstein et al., 2002; Oliehoek et al., 2016), defined by the tuple $\langle \mathcal{N}, \mathcal{S}, \{\mathcal{A}^i\}, P, R, \{\mathcal{O}^i\}, O, \gamma \rangle$, where $\mathcal{N}$ is the set of agents, $\mathcal{S}$ the state space, $\mathcal{A}^i$ the action space of agent $i$, $P$ transition dynamics, $R$ the reward function, $\mathcal{O}^i$ the observation space of agent $i$, and $\gamma$ discount factor. At each timestep $t$, the environment is in global state $S_t \in \mathcal{S}$. Each agent $i \in \mathcal{N}$ receives a local observation $o_t^i \sim O(\cdot | S_t, i)$ and selects an action $a_t^i \in \mathcal{A}^i$. The joint action $A_t = (a_t^i)_{i \in \mathcal{N}} \in \mathcal{A} = \prod_{i \in \mathcal{N}} \mathcal{A}^i$ induces a transition $S_{t+1} \sim P(\cdot \mid S_t, A_t)$ and produces a team reward $r_t = R(S_t, A_t)$.

Each agent $i$ has an action-observation history $\tau_t^i = (o_0^i, a_0^i, \ldots, o_t^i) \in T^i$, and follows a stochastic policy $\pi^i : T^i \to \Delta(\mathcal{A}^i)$, where $T^i$ denotes the set of action-observation histories and $\Delta(\mathcal{A}^i)$ denotes the set of probability distributions on $\mathcal{A}^i$. A joint policy is denoted by $\boldsymbol{\pi} = \prod_{i \in \mathcal{N}} \pi^i$. The objective is to maximize the expected discounted return $J(\boldsymbol{\pi}) = \mathbb{E}_{\boldsymbol{\pi}}[\sum_{t=0}^{\infty} \gamma^t r_t]$. We adopt centralized training with decentralized execution (CTDE): global state information is accessible in training time, whereas in test time agent policies rely solely on local observations.

## 3 PROBLEM STATEMENT

### 3.1 COOPERATION UNDER EXOGENOUS VARIABILITY

We consider cooperative multi-agent decision making under *exogenous variability* by which we mean episode-to-episode changes of the environment that are not directly controlled by the team but do alter the transition dynamics. Specifically, we consider a family of cooperative Dec-POMDPs $\{M_y\}_{y \in \mathcal{Y}}$ indexed by a variation index $y$, which specifies the exogenous regime under which the environment is instantiated. We assume that across different $y$, the state, action and observation spaces $(\mathcal{S}, \{\mathcal{A}^i\}_{i=1}^n, \{\mathcal{O}^i\}_{i=1}^n)$ and the discount factor $\gamma$ are fixed, but the transition kernel, reward function and initial-state distribution may vary. So, these varying components are written by $P^y(S_{t+1} \mid S_t, A_t)$, $R^y(S_t, A_t)$ and $\rho_0^y(S_0)$ to explicitly show their dependence of index $y$. Then, the learning problem is to acquire a joint policy that yields a high return under a training set of Dec-POMDPs $y \in \mathcal{Y}_{train}$ while remaining effective under a set of possibly-shifted Dec-POMDPs $y \in \mathcal{Y}_{test}$ at evaluation. For full generality, we assume that the Dec-POMDP index $y$ affects the transition dynamics at time $t$ through a random variable $Y_t^y$ drawn from $Y_t^y = P_Y(\cdot \mid S_t, y)$. Then, we can write the $y$-dependent transition dynamics as

$$P^y(S_{t+1} \mid S_t, A_t) = P(S_{t+1} \mid S_t, A_t, Y_t^y).$$

In the simplest case where $Y_t^y$ is independent of $S_t$ and has impulse density, then $Y_t^y = y$ is almost surely.

## 3.2 EXOGENOUS DEC-POMDP

To study cooperation under exogenous variability, we now define an *exogenous Dec-POMDP (ED-POMDP)*. In an ED-POMDP, the global state admits a partition $S_t = (S_t^e, S_t^x)$, where $S_t^e$ denotes *endogenous* states controllable by team actions and $S_t^x$ denotes *exogenous* states driven by $Y_t^y$.

**Definition 1 (Team-action $\varepsilon$-exogeneity)** *Let $\Pi_X$ be a projection from $\mathbb{R}^d$ to $\mathbb{R}^{d_x}$, and let $S_t^x = \Pi_X S_t$ and $S_t^e = (\mathbf{I} - \Pi_X)S_t$, where $\mathbf{I}$ denotes the identity matrix. We say $S^x$ is $\varepsilon_X$-exogenous with respect to (w.r.t.) team actions if*

$$I(S_{t+1}^x; A_t \mid S_t, Y_t^y) \leq \varepsilon_X \quad \forall t, \tag{1}$$

*for a small leakage $\varepsilon_X > 0$. We say $S^e$ is $\varepsilon_E$-endogenous if*

$$I(S_{t+1}^e; Y_t^y \mid S_t, A_t) \leq \varepsilon_E \quad \forall t, \tag{2}$$

*for another leakage $\varepsilon_E > 0$. Here, $I(\cdot; \cdot|\cdot)$ denotes conditional mutual information.*

Eq. (1) can be rewritten as $\mathbb{E}_{A_t|S_t, Y_t}\left[D_{KL}\left(P(S_{t+1}^x | S_t, Y_t^y, A_t) \,\|\, P(S_{t+1}^x | S_t, Y_t^y)\right)\right] \leq \varepsilon_X$, and eq. (2) can be rewritten in a similar way. Thus, the KL distance condition yields $P(S_{t+1}^x \mid S_t, Y_t^y, A_t) \approx P(S_{t+1}^x \mid S_t, Y_t^y)$ and $P(S_{t+1}^e \mid S_t, A_t, Y_t^y) \approx P(S_{t+1}^e \mid S_t, A_t)$, leading to the decomposition of the overall transition kernel as follows.

**Lemma 1 (Factorization of Transition Dynamics)** *If $I(S_{t+1}^x; A_t \mid S_t, Y_t^y) = 0$ and $I(S_{t+1}^e; Y_t^t \mid S_t, A_t) = 0$, then the transition kernel factorizes as*

$$P(S_{t+1} \mid S_t, A_t, Y_t^y) = P_e(S_{t+1}^e \mid S_t, A_t) \, P_x(S_{t+1}^x \mid S_t, Y_t^y). \tag{3}$$

Proof. *Note $S_{t+1} = (S_{t+1}^e, S_{t+1}^x)$. Then, the claim follows by the conditional independence and the argument above with $\varepsilon_X = \varepsilon_E = 0$.*

**Causal Influence Beyond the Decision Step** An ED-POMDP enforces decision-time exogeneity. That is, there is no direct effect of $A_t$ on the exogenous transition at time $t$, i.e., on $S_{t+1}^x$. As shown in Fig. 1, however, an action can still have a delayed impact on exogenous state beyond the current decision step: the action first updates the endogenous state ($A_t \to S_{t+1}^e$), and on the next step the exogenous state $S_{t+2}^x$ depends on $(S_{t+1}, Y_{t+1})$, where $S_{t+1} = (S_{t+1}^e, S_{t+1}^x)$. Thus, the causal influence chain $A_t \to S_{t+1}^e \to S_{t+2}^x$ exists. This motivates us to efficiently exploit this causal influence link to our advantage such that indirectly-affected exogenous state by our current action is aligned to help team coordination for high return. In other words, we aim to build a mechanism to cope with exogenous variations by indirectly affect-

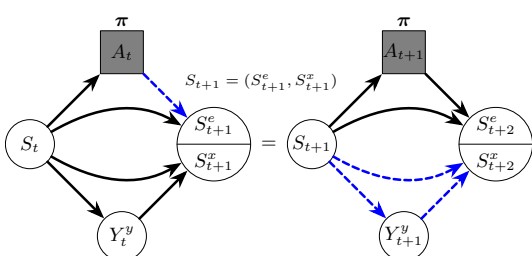

Figure 1: **Two-step causal view (ED-POMDP).** $A_t$ only updates $S_{t+1}^e$ by decision-time exogeneity. At $t + 1$, $S_{t+2}^x$ depends on $S_{t+1}^e$, yielding a delayed effect of $A_t$ on later exogenous variables. The blue arrows highlight this delayed causal pathway from $A_t$ through $S_{t+1}^e$ to the exogenous state $S_{t+2}^x$.

ing exogenous state to align with the team goal, making our policy robust to exogenous variability.

## 3.3 TRAINING AND EVALUATION PROTOCOL

Let $y$ be an episode-level Dec-POMDP index sampled from $\mathcal{Y}_{train}$ during training and from $\mathcal{Y}_{test}$ at evaluation. For an episode with $y$, let $J(\boldsymbol{\pi}; y)$ denote the expected discounted team return under the joint policy $\boldsymbol{\pi}$. The training objective is to maximize expected return for $\mathcal{Y}_{train}$: $\max_{\boldsymbol{\pi} \in \Pi} \mathbb{E}_{y \in \mathcal{Y}_{train}}\left[J(\boldsymbol{\pi}; y)\right]$. Then, robustness and adaptability performance is evaluated on $\mathcal{Y}_{test}$ which may contain unseen exogenous variation not encountered during training. This explicitly quantifies the generalization ability of $\boldsymbol{\pi}$ to remain effective under unseen exogenous variation.

## 4 METHOD

In this section, we present our algorithm *Learning Exogenous Influence for Coordination and Adaptation (LEICA)* for ED-POMDP. LEICA first infers endogenous and exogenous contexts by training predictors of endogenous and exogenous transitions from agent histories. Then, the learned structure is used to quantify how endogenous changes influence future exogenous evolution and team value and to design influence-weighted reward shaping so that policy updates target adaption for coordination.

### 4.1 ENDOGENOUS AND EXOGENOUS INFERENCES

In the ED-POMDP setting, the global state is decomposed as $S_t = (S_t^e, S_t^x)$. At each time $t$, each agent $i$, $i = 1, \cdots, N$, receives an observation $o_t^i$ and the team takes a joint action $A_t = (a_t^1, \ldots, a_t^N)$. To learn the endogenous and exogenous dynamics, we adopt two variational autoencoders (VAEs) for each agent $i$. From the history $\tau_t^i$ of agent $i$, two VAE encoders respectively produce two latent contexts: $z_t^i$ that summarizes factors driving endogenous transitions, and $y_t^i$ that summarizes the exogenous variator. The exogenous context is trained to predict exogenous state transitions, and the endogenous context is trained to predict endogenous state transitions from the perspective of agent $i$. That is, for agent $i$ at time $t$, from the latent variables $(z_t^i, y_t^i)$, the VAE decoders respectively predict the time step $k + 1$ from time step $k$:

$$\hat{S}_{k+1|t}^{e,i} = p_{\psi_z^i}(S_k, a_k^i, z_t^i) \quad \text{and} \quad \hat{S}_{k+1|t}^{x,i} = p_{\psi_y^i}(S_k, y_t^i), \quad k \in \mathcal{K}_t, \tag{4}$$

where $p_{\psi_z^i}$ predicts endogenous transitions, $p_{\psi_y^i}$ predicts exogenous transitions, and $\hat{\cdot}_{k+1|t}$ denotes a predicted quantity for step $k + 1$ made with the context of time $t$. Here, we consider multiple $k$'s from the episode containing time step $t$ for learning efficiency. For this, we sample a small index set $\mathcal{K}_t$ of size $K$ for $k$, reducing computational cost from $\mathcal{O}(T^2)$ to $\mathcal{O}(KT)$ with $T$ being the episode length. The VAEs are trained to minimizes the loss that combines the negative log likelihood terms over $k \in \mathcal{K}_t$ with a KL regularization of the latent contexts toward simple priors. Details and exact loss formulations are given in Appendix A.

The learned components are then used in subsequent learning. The exogenous predictor $p_{\psi_y^i}$ is used to support estimation of the delayed influence of $S_t^e$ on $S_{t+2}^x$, and the contexts $(z_t^i, y_t^i)$ are supplied to the actor and the centralized critic during training. During decentralized execution, the predictors are not used. The actor conditions on statistics of $y_t^i$ to adapt to the current exogenous estimate, and the critic uses $y_t^i$ to form value estimates.

### 4.2 INFLUENCE-BASED COUNTERFACTUAL REWARD DESIGN

Now we use the representations from Section 4.1 to construct learning rewards that favor endogenous changes that influence future exogenous evolution (adaptation) and are sensitive to team value (coordination). Our construction is based on the causal influence link shown in Fig. 1.

We first estimate, for each endogenous state vector coordinate, how strongly the change at time $t$ affects the subsequent exogenous transition.

**Influence Scores and Weight Construction** Consider time step $t + 1$. The exogenous predictor $p_{\psi_y^i}$ for agent $i$ outputs $\hat{S}_{t+2}^{x,i} = p_{\psi_y^i}(S_{t+1}, y_{t+1}^i) = p_{\psi_y^i}(S_{t+1}^e, S_{t+1}^x, y_{t+1}^i)$. Here, the upper index $i$ for $\hat{S}_{t+2}^{x,i}$ indicates agent $i$'s predictor, but all $\hat{S}_{t+2}^{x,i}$, $i = 1, \cdots, N$ estimate the same global exogenous state $S_{t+2}^x$. Then, we compute the Jacobian of this prediction function w.r.t. the endogenous state at each time $t + 1$:

$$J_{xe}^i = \frac{\partial \hat{S}_{t+2}^{x,i}}{\partial S_{t+1}^e} = \frac{\partial \, p_{\psi_y^i}(S_{t+1}^e, S_{t+1}^x, y_t^i)}{\partial S_{t+1}^e} \in \mathbb{R}^{d_x \times d_e}. \tag{5}$$

This quantity measures the local effect $S_{t+1}^e \to S_{t+2}^x$ under the current exogenous context $y_{t+1}^i$. Then, for each endogenous dimension $j$, we define its influence on the next exogenous state by the $\ell_2$ norm of the $j$-th column of the Jacobian matrix:

$$c_j^x = \frac{1}{N} \sum_{i=1}^N \left\| J_{xe}^i[:, j] \right\|_2, \quad j = 1, \ldots, d_e, \tag{6}$$

where the endogenous state vector size is denoted by $d_e$.

To align this influence with return, we also score how the current endogenous state affects value. That is, let $V_\phi$ be the centralized value with input $S_{t+1} = (S_{t+1}^e, S_{t+1}^x)$. We compute the gradient w.r.t. the current endogenous state for each time step $t + 1$: $g_e = \nabla_{S_{t+1}^e} V_\phi(S_{t+1})$, and define the value-alignment score for the $j$-th coordinate of the endogenous state vector as follows:

$$c_j^v = \left| g_{e,j} \right|, \quad j = 1, \ldots, d_e. \tag{7}$$

We aggregate vectors $c^x$ and $c^v$ of eqs. (6) and 7 over all timesteps $t$'s in the minibatch and over all individual predictors, and standardize each across dimensions to zero mean and unit variance, giving $\bar{c}^x$ and $\bar{c}^v \in \mathbb{R}^{d_e}$, respectively. Then, we combine them with a parameter $\alpha \in [0, 1]$ and construct a positive weighting vector $\tilde{w}$:

$$c = \alpha\, \bar{c}^x + (1 - \alpha)\, \bar{c}^v \ \text{ and } \ \tilde{w} = \mathrm{softmax}(c/\tau), \tag{8}$$

where $\alpha$ sets a trade-off between exogenous-influence ($\alpha = 1$) and value-sensitivity ($\alpha = 0$), and $\tau > 0$ controls the sharpness. Finally, we apply an exponential moving-average filter to $\tilde{w}$ for temporal stability, yielding the weight $w \leftarrow (1 - \beta)w + \beta\tilde{w}$ with $\beta \in (0, 1]$. Note that filtered $w$ carries the time-spanning (not instantaneous) information about each coordinate of the endogenous state vector's sensitivity to value and total influence on the next exogenous state. We will use $w$ to weight the counterfactual endogenous changes to obtain an intrinsic reward below.

For the trade-off parameter $\alpha$, we use exponentially decaying $\alpha$, which starts with high $\alpha$ early in learning when value estimation is unstable and requires more adaptation to opponents, and gradually decreases to emphasize on value as learning progresses.

**Influence-weighted Intrinsic Reward**  Now consider time step $t$. Here, we quantify the impact of current action on endogenous state change in the next time step $t + 1$. We have the estimate of the endogenous state at $t + 1$ of action $a_t^i$ from the predictor $p_{\psi_z^i}$ as $\hat{S}_{t+1}^{e,i}(a_t^i) = p_{\psi_z^i}(S_t^e, S_t^x, a_t^i, z_t^i)$ for agent $i$, where we hide the dependence on $S_t$, $z_t^i$ for the estimate for notational simplicity. To assess the relativity performance of $a_t^i$ over different actions, we consider a baseline action $\bar{a}_t^i$ for agent $i$ and obtain the corresponding endogenous state estimate from the predictor $p_{\psi^i}$, i.e., $\hat{S}_{t+1}^{e,i}(\bar{a}_t^i)$. Then, we apply counterfactual reasoning to measure action $a_t^i$'s performance against the average behavior:

$$\Delta e_t^i = \left| \hat{S}_{t+1}^{e,i}(a_t^i) - S_t^e \right| - \mathbb{E}_{\bar{a}_t^i \sim \pi^i}\left[ \left| \hat{S}_{t+1}^{e,i}(\bar{a}_t^i) - S_t^e \right| \right] \in \mathbb{R}^{d_e}, \tag{9}$$

where $|\cdot|$ denotes element-wise absolute value taking. This quantity captures action $a_t^i$'s contribution to coordinate-wise endogenous state change while holding the team inference $z_t^i$ fixed.

Now, based on the causal influence link $A_t \rightarrow S_{t+1}^e \rightarrow S_{t+2}^x$ shown in Fig. 1, we can compute action $a_t^i$'s impact on the exogenous state at $t + 2$ and value sensitivity at $t + 1$ by combining $\Delta e_t^i$ and $w$, where $w \in \Delta^{d_e - 1}$ is the influence weights over the endogenous coordinates obtained above. Then, we define the intrinsic reward by scaling the counterfactual change with these weights:

$$r_t^{\mathrm{int},i} = \langle w, \Delta e_t^i \rangle, \tag{10}$$

where the gradient is stopped to prevent the intrinsic reward signal from updating the predictor network. The intrinsic reward is added to the extrinsic reward $r_t^{\mathrm{ext}}$ from the environment, yielding the final reward:

$$\tilde{r}_t^i = r_t^{\mathrm{ext}} + \lambda r_t^{\mathrm{int},i}. \tag{11}$$

where $\lambda$ is the weighting factor for reward shaping.

The proposed method, LEICA, is composed of the context inference structure (4) and the reward shaping (11) to solve MARL problems with exogenous variability under the ED-POMDP framework. Actors are trained with the clipped surrogate objective from MAPPO (Yu et al., 2022) using the individual shaped rewards $\tilde{r}_t^i$, $i = 1, \cdots, N$, whereas the centralized critic regresses returns based on the average $\tilde{r}_t = \frac{1}{N} \sum_{i=1}^{N} \tilde{r}_t^i$. Note that actor-critic gradients are not backpropagated into the predictors and the representation modules are updated solely by the losses in Section 4.1. For implementation details and the overall algorithm overview, please refer to Appendix A.

## 5 RELATED WORKS

**MARL under CTDE**   CTDE is the standard paradigm for cooperative MARL with partial oberv-
ability: a centralized critic stabilizes learning while decentralized actors execute from local obser-
vations. Major actor-critic variants include MADDPG and MAA2C (Lowe et al., 2017; Papoudakis
et al., 2020), on-policy MAPPO and IPPO (Yu et al., 2022; De Witt et al., 2020), and theory-based
trust-region methods HATRPO and HAPPO with heterogeneous-agent extensions (Kuba et al., 2021;
Zhong et al., 2024). For value-based methods, VDN and QMIX implement additive and monotonic
mixing based on Individual-Global-Maximum (IGM) principle (Sunehag et al., 2017; Rashid et al.,
2018), and many subsequent studies investigate the value decomposition structure of MARL (Son
et al., 2019; Wang et al., 2020; Rashid et al., 2020; Yang et al., 2020; Zheng et al., 2021; Son et al.,
2022).

**Influence-based Coordination**   Influence-based methods, which provide intrinsic rewards based
on the degree of influence of other players and the environment, have been widely used to encour-
age team coordination and to adapt to opponents and environments. Some prior works have used
message to influence other players (Strouse et al., 2018; Foerster et al., 2016), or have proposed
metrics to quantify causal influence (Lowe et al., 2019). These approaches aim to alter other agents'
strategies through messages, whereas our work does not employ a separate communication channel.
Instead, we quantify the influence of an agent's actions on the state and use it as an intrinsic reward.

Alternatively, influence can arise directly from agents' actions. Some approaches used influence to
encourage exploration, particularly in sparse-reward settings (Wang et al., 2019; Ma et al., 2022; Liu
et al., 2023). In these works, the primary role of influence is to facilitate exploration. In contrast,
in our case the purpose of influence is twofold: adaptation to unseen variations and coordination.
Opponent-aware learning methods aim not only to optimize an agent's own value but also to di-
rectly influence opponents' policies or value functions, thereby manipulating them toward helpful
behaviors (Foerster et al., 2017; Zhao et al., 2022; Aghajohari et al., 2024; Duque et al., 2024). These
approaches exploit the fact that opponents are concurrently co-learning with the agent, so one's mes-
sages or actions can directly affect the opponents' value functions and, consequently, their learning
process. In contrast, in our setting the opponents are not learning but are diverse yet externally
specified, and thus the agent's actions do not directly influence their policies.

**Exogenous Structure and Context Latent**   In this paper, we investigate multi-agent situations
where the part of the environment is transitioned exogenously, beyond the team's direct control.
Relevant studies on single-agent RL formalizes this uncontrollable variation as explicit exogenous
processes to improve sample efficiency (Efroni et al., 2022; Wan et al., 2024). In contrast, we allow
the exogenous process to vary, requiring adaptation to this variability. A related line of meta-RL
studies encode the latent context from interactions and train it with auxiliary objectives for rewards
or state dynamics (Duan et al., 2016; Rakelly et al., 2019; Zintgraf et al., 2019). Comparable con-
cerns arise in multi-agent settings: meta-learning perspective in decentralized formulations (Kayaalp
et al., 2022) and game-theoretic analysis (Mao et al., 2023). Unlike these lines focused on learning
individual agent, we study cooperative teams how to adapt to exogenous shifts while maintaining
coordination.

## 6 EXPERIMENTS

### 6.1 SETUP

We evaluate LEICA on SMAX (Rutherford et al., 2024), which is a JAX-based implementation of
SMAC (Samvelyan et al., 2019). SMAX runs natively on GPU, delivering high-throughput rollouts
and freeing us from the StarCraft II engine, which makes systematic environment modifications
straightforward. This flexibility is essential for exogenous variability, as it enables to alter policies
of opponent units.

We evaluate on five cooperative scenarios in SMAX with variability of opponent strategy:
`5m_vs_6m`, `10m_vs_11m`, `3s5z_vs_3s6z`, `6h_vs_9z`, and `3s_vs_6z`. The first three are standard
SMAX tasks already known to be difficult, while the last two are *extreme-hard* variants obtained by
increasing the enemy counts of the original `6h_vs_8z` and `3s_vs_5z` scenarios, respectively. To

induce exogenous variation, we define a strategic space with five options for the enemy in the SMAX environment, resulting in a total of 63 distinct strategies. Beyond the strategy, we vary the initial state based on the initial distance between enemies and allies: *close*, *normal*, and *far*. This leads to 189 exogenous configurations overall.

Based on the exogenous configurations, we evaluated LEICA across three scales: **small** (training 3, testing 3) fixes the initial distance to *normal* and employs relatively simple strategies. **medium** (training 10, testing 5) and **large** (training 30, testing 10) encompass a broader range of strategies and present more challenging generalization tasks, including initial distance variations. Only the exogenous configurations in training set are used during the training, and the performance is evaluated separately for adaptability to the seen training set and generalizability to the unseen test set. For detailed definitions, selection procedures of strategic axes and the ED-POMDP state partition, refer to Appendix B.

We compared LEICA against a comprehensive set of strong baselines: MAPPO (Yu et al., 2022), QMIX (Rashid et al., 2018), LAIES (Liu et al., 2023), SHAQ (Wang et al., 2022), as well as two robust MARL baselines, M3DDPG (Li et al., 2019), a robustness-enhanced variant of MADDPG (Lowe et al., 2017), and M3PPO, our MAPPO-based extension of M3DDPG. MAPPO and QMIX represent standard CTDE actor-critic and value-decomposition methods. LAIES augments QMIX with state-based intrinsic rewards, and SHAQ incorporates Shapley-value-based credit assignment. Our implementations use JAX/Flax (Bradbury et al., 2018; Heek et al., 2024). Hyperparameters and training details are provided in Appendix C. All methods are trained for 10M environment steps under identical settings. We report mean performance over 8 seeds with standard error of the mean. Policies are checkpointed every 0.5M steps and evaluated over multiple test episodes per checkpoint. Win rate statistics and learning curves are reported for overall performance and sample efficiency, with per-task results given in Appendix F. For details on computational cost and wall-time efficiency, please refer to Appendix B.4.

Table 1: **Main results on SMAX with exogenous variation.** The table shows the average win rate performance (%) with standard error of mean over 8 seeds. Each method has Train/Test (seen/unseen) subcolumns. Regimes: small, medium, and large. The best performance on unseen test set per row is bold and the best performance on training set per row is underlined.

| Task | Regime | MAPPO Train | MAPPO Test | QMIX Train | QMIX Test | LAIES Train | LAIES Test | SHAQ Train | SHAQ Test | M3DDPG Train | M3DDPG Test | M3PPO Train | M3PPO Test | LEICA Train | LEICA Test |
|---|---|---|---|---|---|---|---|---|---|---|---|---|---|---|---|
| 6h_vs_9z | small | 50.8(7.6) | 50.0(9.9) | 68.9(3.9) | 66.9(3.2) | 60.9(7.3) | 61.0(7.1) | 7.7(1.2) | 6.2(1.1) | 0.7(0.3) | 1.6(0.5) | 38.7(9.2) | 41.0(9.8) | 83.1(2.6) | **84.6(0.8)** |
| | medium | 57.6(3.4) | 22.6(4.8) | 35.3(1.2) | 0.9(0.3) | 52.4(2.4) | 7.7(0.8) | 20.6(2.4) | 14.1(1.8) | 24.0(4.1) | 0.5(0.5) | 39.2(0.4) | 0.0(0.0) | 77.4(3.3) | **56.9(2.7)** |
| | large | 58.4(2.5) | 21.0(2.8) | 40.9(0.8) | 5.3(0.4) | 51.3(1.7) | 9.9(0.9) | 31.9(1.6) | 23.7(1.0) | 27.6(4.9) | 3.0(1.0) | 47.1(0.9) | 9.3(0.7) | 66.9(1.4) | **38.6(0.9)** |
| 5m_vs_6m | small | 62.4(3.7) | 62.3(0.8) | 61.8(2.4) | 61.1(1.7) | 34.8(9.4) | 33.3(10.3) | 0.5(0.1) | 0.3(0.1) | 28.9(7.4) | 26.9(7.3) | 51.1(13.7) | 41.2(11.9) | 86.1(2.1) | **73.1(3.6)** |
| | medium | 44.4(4.4) | 8.3(5.3) | 73.1(1.9) | 11.6(0.5) | 76.2(1.4) | 11.4(0.7) | 23.3(0.4) | 17.9(0.9) | 18.0(1.6) | 0.7(0.7) | 38.1(0.5) | 0.1(0.0) | 61.6(7.3) | **34.6(6.0)** |
| | large | 48.2(0.8) | 9.7(0.7) | 50.0(1.9) | 14.5(2.7) | 58.6(3.3) | 13.6(1.7) | 36.2(0.8) | **30.9(0.5)** | 4.9(1.4) | 0.0(0.0) | 46.4(0.6) | 9.6(0.3) | 53.4(4.5) | 20.9(7.3) |
| 10m_vs_11m | small | 92.3(1.3) | 67.8(0.5) | 54.1(6.6) | 51.6(7.0) | 60.6(10.2) | 55.0(9.0) | 2.9(0.8) | 1.9(0.4) | 5.1(1.2) | 5.1(1.2) | 38.2(8.2) | 14.1(7.4) | 96.4(1.1) | **81.6(3.3)** |
| | medium | 60.4(6.3) | 25.6(9.6) | 62.9(7.4) | 8.3(2.4) | 70.7(6.0) | 13.0(3.0) | 28.4(0.6) | 23.1(0.9) | 15.0(3.2) | 0.1(0.1) | 13.0(1.1) | 0.1(0.1) | 86.0(1.4) | **65.3(1.2)** |
| | large | 56.1(4.1) | 20.7(5.9) | 48.0(1.8) | 10.1(1.1) | 67.6(4.5) | 25.0(7.2) | 36.2(1.4) | 29.3(1.0) | 14.5(2.3) | 0.2(0.2) | 45.8(4.0) | 9.0(1.4) | 67.8(5.0) | **37.9(7.6)** |
| 3s_vs_6z | small | 98.4(1.1) | 98.6(0.6) | 82.1(7.9) | 81.6(7.8) | 19.0(13.1) | 19.5(13.3) | 0.0(0.0) | 0.0(0.0) | 0.0(0.0) | 0.0(0.0) | 73.0(12.1) | 74.3(11.6) | 99.9(0.1) | **99.9(0.1)** |
| | medium | 65.5(9.5) | 41.6(16.2) | 89.3(1.5) | 60.1(3.9) | 45.9(3.5) | 4.5(3.0) | 24.5(0.6) | 11.0(0.9) | 13.6(2.4) | 0.0(0.0) | 50.4(4.8) | 7.8(4.5) | 99.7(0.1) | **99.5(0.2)** |
| | large | 57.0(6.2) | 25.1(10.7) | 91.3(2.1) | 83.5(3.5) | 51.7(1.5) | 8.6(1.5) | 38.2(3.4) | 25.4(2.7) | 16.2(3.2) | 2.1(0.9) | 48.7(2.6) | 11.4(1.7) | 99.7(0.1) | **99.8(0.1)** |
| 3s5z_vs_3s6z | small | 69.7(10.2) | 54.0(8.9) | 51.6(2.3) | 46.3(2.8) | 48.2(3.9) | 35.6(6.0) | 1.0(0.4) | 0.4(0.2) | 0.0(0.0) | 0.0(0.0) | 9.4(3.5) | 2.3(1.1) | 95.7(1.5) | **80.8(4.0)** |
| | medium | 40.5(0.8) | 0.7(0.7) | 30.9(1.8) | 3.4(0.5) | 53.5(2.8) | 12.3(3.0) | 13.0(1.2) | 8.8(1.0) | 9.8(0.9) | 0.0(0.0) | 18.0(3.0) | 1.1(1.1) | 55.5(4.9) | **26.3(8.8)** |
| | large | 47.8(0.7) | 9.8(0.7) | 30.7(2.2) | 3.9(0.7) | 56.8(1.9) | 12.8(2.0) | 23.1(1.8) | 13.7(1.0) | 11.3(1.5) | 0.0(0.0) | 44.6(1.9) | 8.1(0.8) | 60.5(4.1) | **30.3(7.6)** |
| Overall avg. (all 15) | | 60.6 | 34.5 | 58.1 | 33.9 | 53.9 | 21.5 | 19.2 | 13.8 | 12.6 | 2.7 | 40.1 | 15.3 | 79.3 | **62.0** |

## 6.2 MAIN RESULTS

Table 1 summarizes the SMAX performance in 15 pairs of task-regimes (5 each for the small, medium, and large regimes) with exogenous perturbations applied. The left column for each algorithm shows the evaluation performance on the training set variations, while the right column shows performance on the untrained test set. LEICA achieved the highest performance on both the learned and unseen sets, and also demonstrated superior performance on most individual tasks. The training-test gap for LEICA was 17.3%p, lower than MAPPO's 26.2%p, QMIX's 24.2%p, and LAIES's 32.4%p. This demonstrates excellent unseen generalization capability while maintaining the highest performance on the train set. SHAQ, on the other hand, shows significantly lower performance across most tasks, with the exception of a few specific ones (5m_vs_6m large).

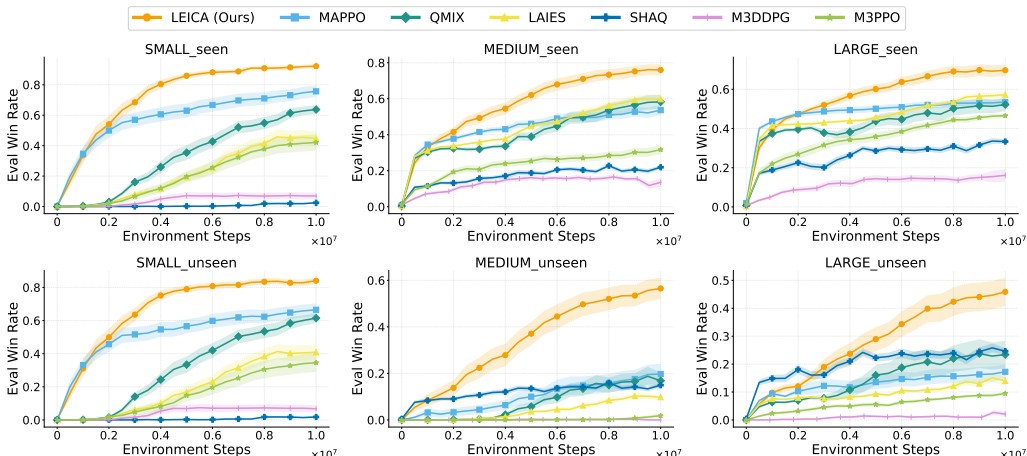

Figure 2: **Main results on SMAX with opponent strategy variation.** Aggregated learning curves of win rate performance for LEICA and baselines across small/medium/large settings and seen/unseen exogenous regimes. The upper three figures show performance graphs for the seen training set, while the bottom three figures show performance for the unseen test set.

We also evaluate two robust MARL baselines, M3DDPG and M3PPO, to examine whether max-min robustness helps under exogenous variation. M3DDPG shows consistently poor performance, in line with the known weaknesses of MADDPG in cooperative tasks (Papoudakis et al., 2020). To rule out backbone issues, we test M3PPO, a PPO-based version of M3DDPG, but it still underperforms vanilla MAPPO. These results indicate that traditional max-min robust MARL methods do not effectively handle exogenous variability.

The learning curve further highlights LEICA's superiority. Fig. 2 shows the aggregated win-rate performance across all three scales averaged over tasks. The upper three figures represent performance evaluated on seen train set, while the bottom three figures represent performance evaluated on unseen test set. LEICA demonstrates outstanding performance across all six results, showing significant performance gap over the other algorithms at medium and large scales where their test performance collapses.

### 6.3 VISUALIZATION OF INFLUENCE WEIGHT

To further understand how LEICA adjusts its intrinsic reward during training, we visualize the influence weights $w_j$ for each endogenous state coordinate on the `3s_vs_6z` scenario. Figure 3 shows how intrinsic rewards progressively shift focus across different endogenous dimensions as learning progresses.

Early in training, the influence weight concentrates almost entirely on ally position dimensions (`ally0_X`, `ally1_X`, `ally2_X`). At this stage, the value function is still unstable due to the large diversity of opponent behaviors across the exogenous family, and LEICA thus relies more on intrinsic reward. By emphasizing controllable positional changes, the agent learns how its actions modulate exogenous dynamics, which helps the policy and value function effectively distinguish and interpret diverse opponents. Here, the reason only changes along the $x$-axis are considered is due to the map's geometric structure: the $y$-axis is narrow, and both teams spawn at similar $y$-coordinates but are primarily separated along the $x$-axis, making horizontal movement far more impactful. Therefore, most movement naturally occurs along the $x$-direction.

Later in training, as the value function becomes more accurate and distribution-aware, the influence shifts dramatically toward enemy health dimensions (`enemy0_HP − enemy5_HP`). This reflects the intended effect of the exponentially decaying $\alpha$: once the agent has learned the controllable mechanisms that structure exogenous variability, LEICA progressively aligns its weights to the extrinsic objective. The policy therefore transitions from exploring exogenous dynamics and resolving am-

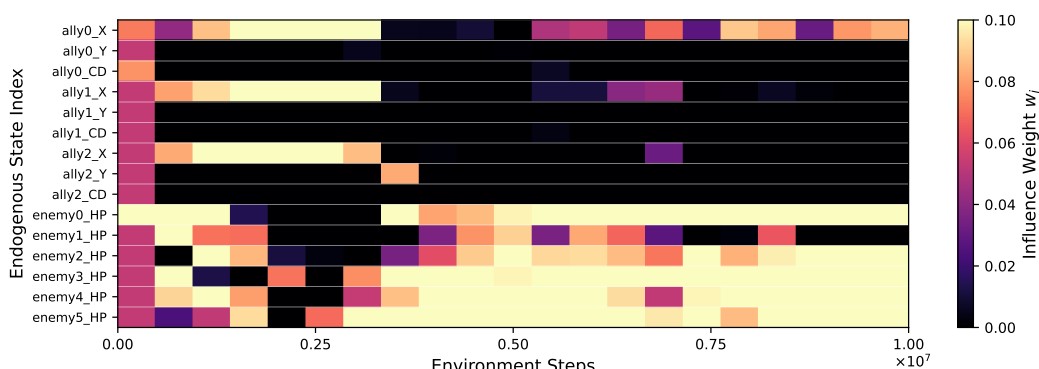

Figure 3: Evolution of influence weights $w_j$ on `3s_vs_6z`. In the early stages of training, LEICA assigns high weight to the ally's $x$-position coordinate, enabling the policy to learn how controllable movements regulate various exogenous dynamics. As training progresses, weight shifts toward the enemy's health dimension, reflecting the intent of exponentially decaying $\alpha$: intrinsic shaping guides initial structural learning of exogenous variability, then gradually yields to extrinsic goals as the agent becomes proficient.

biguity to exploiting them, focusing on endogenous factors that directly contribute to maximizing returns.

### 6.4 ABLATIONS AND COMPONENT EVALUATION

We isolate the contribution of each design component with controlled ablation study:

**Balance of Adaptation and Coordination**  We compare four variants that differ in how the influence coefficient $\alpha$ is set when forming the influence weights $w$ (Sec. 4.1): (i) $\alpha=1$ (pure exogenous, value alignment removed), (ii) $\alpha=0$ (pure value alignment), (iii) a constant ($\alpha=0.5$), and (iv) an exponentially decaying schedule, which is our default setting. As shown in Fig. 4a, the decaying schedule yields the best performance, followed by the constant $\alpha$. Pushing the weighting to either consistently degrades the performance. This pattern matches our intent: early in training, emphasizing exogenous influence helps discover controllable coordinates that reliably impact opponent-driven dynamics when value estimates are noisy, while gradually shifting weight toward value alignment exploits those coordinates for return as the critic stabilizes.

**Reward Validity**  We test whether adaptation is possible using only learned context without influence-based rewards, and whether counterfactual reasoning is necessary. We compare: (a) *no reward*, where intrinsic rewards are removed while all other components remain the same. The agent and critic can still condition on the learned context; (b) *no counterfactual*, which replaces the $\Delta_t^i$ with a simple one-step size $|\hat{S}_{t+1}^{e,i} - S_t^e|$ without counterfactual baseline; (c) the original *LEICA* with counterfactual reward. As shown in Fig. 4b, the context-only variant learns reasonable adjustments in seen environments but exhibits limited adaptability in unseen opponent strategy, with a notably large train-test performance gap. The variant without counterfactual reasoning showed even more consistent performance degradation than in reward-free scenarios. Notably, its degraded performance even in the training environment demonstrates that variance from intrinsic rewards without counterfactual design significantly hinders learning. In contrast, LEICA consistently maintains higher performance, suggesting that our intrinsic reward structure is key to transforming contextual inference into a stable credit signal rather than a generic task detector.

**Effect of State Separation**  We study the benefits of endogenous-exogenous separation by training a single predictor that reconstructs the full state. We report four settings: (a) Full State Reconstruction with same reward structure of LEICA; (b) Full State, $\alpha=0$ (value sensitivity only); (c) Full State, $\alpha=1$ (exogenous influence only); (d) Full State, no intrinsic reward (actor/critic still receives context). The full state reconstruction variant shows lower performance than LEICA in both seen

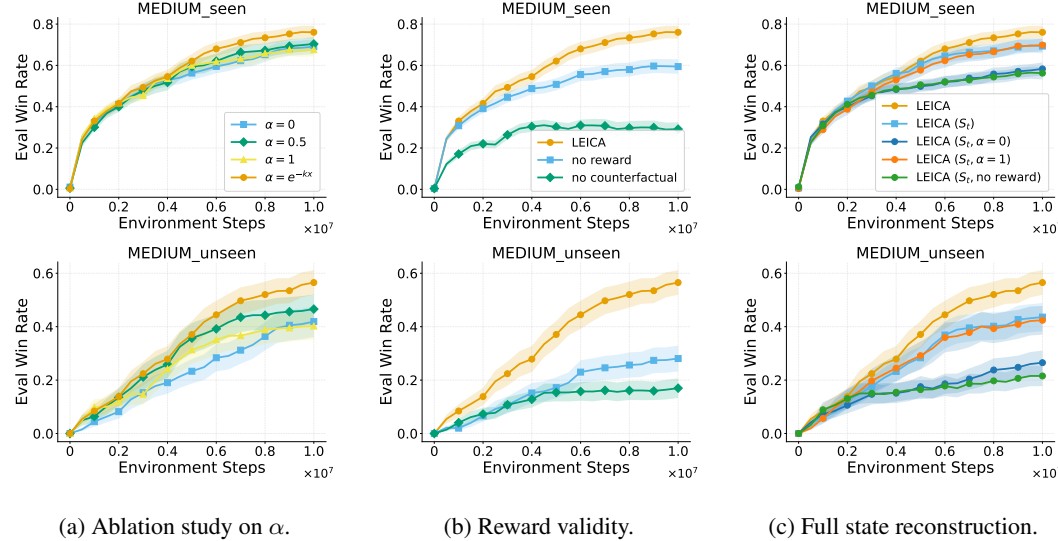

(a) Ablation study on $\alpha$.     (b) Reward validity.     (c) Full state reconstruction.

Figure 4: Ablation study and component evaluation for LEICA framework. The upper three figures show performance graphs for the seen training configurations, while the bottom three figures show performance for the unseen test configurations, respectively. Note that the graphs at the bottom use the same legend as the graphs above them.

and unseen set. While performance is similar in the early stages of training, it converges to a lower level as training progresses. This can be attributed to its inability to learn more complex contexts. According to Fig. 4a, pushing the weight to either extreme degrades LEICA's performance. In contrast, for the full state reconstruction variant, performance degrades significantly only when $\alpha=0$, while it remains similar when $\alpha=1$. It is noteworthy that an intrinsic reward that learns the transition dynamics and strongly influences them is beneficial. The full state variant without reward achieves the lowest performance on both seen and unseen set. These results suggest that (i) endogenous-exogenous decomposition is necessary to obtain informative influence weights, and (ii) an intrinsic reward formed by weights reflecting the degree of state changes facilitate learning.

## 7 CONCLUSION AND FUTURE WORK

In this paper, we reframe cooperative MARL under exogenous variability under ED-POMDP and propose **LEICA**, a CTDE-compatible method that unifies two ideas in a single training reward. First, we learn to infer how exogenous dynamics respond under different variators. Second, we shape policy updates by weighting counterfactual endogenous changes with data-driven influence weights that trade off adaptation and coordination. By integrating these signals, our method enables the learned weights to serve as a readable credit map for controllable state. On SMAX tasks including extreme hard ones with various train/test variants, LEICA improves the performance and speeds early-episode adaptation, with ablations confirming the contribution of each component.

Two directions for future work derived from the limitations of this work are as follows. (i) Learning the partition: rather than assuming an endogenous-exogenous split, a learnable partition can be implemented through learnable masks over latent coordinates, allowing ED-POMDP to be applied without manual feature grouping. This will make influence estimates remain meaningful beyond hand-crafted decompositions. (ii) Beyond local, one-step sensitivities: develop multi-step influence estimators and more principled counterfactual baselines to better align shaping with long-horizon objectives under distribution shift.

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

# A APPENDIX: METHOD DETAILS

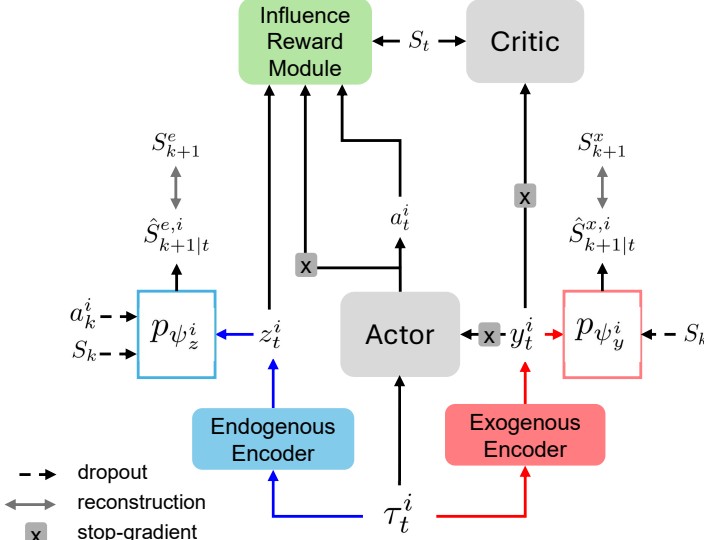

Figure 5: **LEICA overview.** Under CTDE, observations pass through endogenous and exogenous encoders to train one-step predictors. The influence-reward module computes influence weights over endogenous coordinates from exogenous sensitivity and value-aligned scores and forms an intrinsic reward by weighting the magnitude of counterfactual endogenous change.

## A.1 DETAILS REGARDING THE EXOGENOUS AND ENDOGENOUS INFERENCE

For each agent $i$ and time $t$, we infer two stochastic contexts with reparameterization,

$$q_\phi\big(z_t^i \mid \tau_{\leq t}^i\big) = \mathcal{N}\big(\mu_{z,t}^i, \mathrm{diag}(\sigma_{z,t}^{i\,2})\big), \qquad q_\phi\big(y_t^i \mid \tau_{\leq t}^i\big) = \mathcal{N}\big(\mu_{y,t}^i, \mathrm{diag}(\sigma_{y,t}^{i\,2})\big),$$

where the history encoder for contexts is a shared GRU (Chung et al., 2014) over the agent's trajectory $\tau_{\leq t}^i$ (observations/actions, and agent identity is provided via one-hot), and $z_t^i, y_t^i$ are sampled via the standard reparameterization trick. We use filtered priors in the style of variBAD (Zintgraf et al., 2019):

$$p(z_t^i) = \begin{cases} \mathcal{N}(0, I), & t = 0, \\ q_\phi(z_{t-1}^i \mid \tau_{\leq t-1}^i), & t > 0, \end{cases} \qquad p(y_t^i) = \begin{cases} \mathcal{N}(0, I), & t = 0, \\ q_\phi(y_{t-1}^i \mid \tau_{\leq t-1}^i), & t > 0. \end{cases} \tag{12}$$

The context GRU is not shared with the actor or critic encoders. The two contexts share the same GRU and branch into separate heads for each $\mu$ and $\sigma$. With $(z_t^i, y_t^i)$, we train two one-step predictors on multiple indices $k$ sampled from the same episode:

$$\hat{S}_{k+1|t}^{e,i} = p_{\psi_z^i}\big(S_k, a_k^i, z_t^i\big), \qquad \hat{S}_{k+1|t}^{x,i} = p_{\psi_y^i}\big(S_k, y_t^i\big), \qquad k \in \mathcal{K}_t \subseteq \{0, \dots T\}.$$

Here $S_k = (S_k^e, S_k^x)$ includes both endogenous and exogenous coordinates for inputs to both predictors. To respect decision-time exogeneity, the exogenous predictor does not take actions as input. The endogenous predictor conditions on the individual action $a_k^i$. We subsample a fixed number $|\mathcal{K}_t| = K$ of indices uniformly at random for efficiency, yielding $\mathcal{O}(KT)$ cost per episode instead of $\mathcal{O}(T^2)$. During decoder training we apply dropout $p = 0.5$ to the predictor MLPs.

**Predictive Objective and KL Regularization** For each anchor $(i, t)$ we optimize two separate ELBOs. One for the endogenous context $z_t^i$ and one for the exogenous context $y_t^i$.

Endogenous (for $z_t^i$ and $p_{\psi_z}$):

$$\mathcal{L}_{\mathrm{ELBO},z}^{i,t} = \mathbb{E}_{q_\phi(z_t^i)}\left[\sum_{k \in \mathcal{K}_t} \log p_{\psi_z}\big(S_{k+1}^e \mid S_k, a_k^i, z_t^i\big)\right] - \beta_z\,\mathrm{KL}\big(q_\phi(z_t^i) \,\|\, p(z_t^i)\big),$$

Exogenous (for $y_t^i$ and $p_{\psi_y}$):

$$\mathcal{L}_{\mathrm{ELBO},y}^{i,t} = \mathbb{E}_{q_\phi(y_t^i)}\left[\sum_{k\in\mathcal{K}_t}\log p_{\psi_y}(S_{k+1}^x\mid S_k, y_t^i)\right] - \beta_y\,\mathrm{KL}(q_\phi(y_t^i)\,\|\,p(y_t^i)),$$

with filtered priors from Eq. 12.

Practically, we use reconstruction MSE loss for predictive object with reconstruction scale parameter $\lambda_{recon}$, and we use a single reparameterized sample from each posterior to estimate the expectations. The total objective maximized is

$$\mathcal{L}_{\mathrm{ELBO}} = \lambda_{\mathrm{ELBO}}\sum_{t=0}^{T}\sum_{i=1}^{N}(\mathcal{L}_{\mathrm{ELBO},z}^{i,t} + \mathcal{L}_{\mathrm{ELBO},y}^{i,t}).$$

Context encoders and predictors are updated only by $\mathcal{L}_{\mathrm{ELBO}}$ above. As in the main text, gradient is stopped to the influence weights $w$ and to the counterfactual endogenous change $\Delta e_t^i$ when forming intrinsic rewards, ensuring policy/critic gradients do not flow into the representation modules. The actor/critic stacks use their own encoders, without parameter sharing with the context GRU.

For each episode, we iterate anchors $t = 0, \ldots, T-1$ (and agents $i$), draw $K$ indices per anchor, and accumulate $\mathcal{L}^{i,t}$. Vectorized batching over anchors and $k\in\mathcal{K}_t$ yields linear complexity in $KT$ per episode. We found $K$ small ($K\in\{4,8\}$) sufficient in practice.

### A.2 DETAILS REGARDING THE INFLUENCE-BASED REWARD

**Counterfactual Baseline**  Counterfactual baselines are widely used to reduce variance and to provide local credit assignment in policy gradient methods. In the single-agent case, adding a state-dependent baseline $b(S_t)$ leaves the gradient unbiased since

$$\mathbb{E}_\pi[\nabla_\theta\log\pi_\theta(A_t\mid O_t)\,b(S_t)] = \mathbb{E}_\pi\left[b(S_t)\,\nabla_\theta\sum_a\pi_\theta(a\mid O_t)\right] = 0,$$

so one may replace returns with advantages $G_t - b(S_t)$ without bias (Williams, 1992).

In cooperative multi-agent settings under CTDE, the counterfactual idea is used to isolate the marginal effect of a single agent while marginalizing over its own action. Practically, this choice provides variance reduction and agent-level credit signals, and has been widely adopted in related multi-agent policy gradient methods (Foerster et al., 2018; Shao et al., 2023). The canonical example is COMA (Foerster et al., 2018), which defines the agent-wise counterfactual advantage

$$A_{\mathrm{COMA}}^i(S_t, A_t) = Q(S_t, A_t) - \sum_{a'^i}\pi^i(a'^i\mid o_t^i)\,Q\big(S_t, (a'^i, A_t^{-i})\big),$$

holding other agents fixed at $A_t^{-i}$.

In LEICA, the same principle is applied not to $Q$ values but to predicted endogenous transitions, inspired by recent research (Park et al., 2025). The baseline action for agent $i$ is defined as a policy expectation at time $t$,

$$\hat{S}_{t+1}^{e,i}(\bar{a}_t^i) = \mathbb{E}_{\bar{a}_t^i\sim\pi^i(\cdot\mid o_t^i)}\big[p_{\psi_z^i}(S_t, \bar{a}_t^i, z_t^i)\big],$$

which yields a counterfactual change of the endogenous coordinates when contrasted against the realized action (see Sec. 4.2). This substitution preserves the spirit of counterfactual baselines, marginalizing the agent's own action under its current policy, while tailoring the signal to LEICA's objective of weighting endogenous changes by their downstream exogenous influence and value sensitivity.

Practically, weights are updated once per actor-critic update using the same time-batch and are decoupled from PPO epochs. Predictors and encoders are trained only by their predictive losses. During execution, predictors are not used while context statistics may be used.

# B   APPENDIX: ENVIRONMENT AND EVALUATION PROTOCOL DETAILS

## B.1   PLATFORM AND SCENARIOS

We use JaxMARL (Rutherford et al., 2024) for SMAX environments.[1] Table 2 lists the five scenarios and episode limits.

| Name | Ally Team | Enemy Team | Max Steps |
|------|-----------|------------|-----------|
| 5m_vs_6m | 5 Marines | 6 Marines | 70 |
| 10m_vs_11m | 10 Marines | 11 Marines | 150 |
| 3s5z_vs_3s6z | 3 Stalkers, 5 Zealots | 3 Stalkers, 6 Zealots | 170 |
| 6h_vs_9z | 6 Hydralisks | 9 Zealots | 150 |
| 3s_vs_6z | 3 Stalkers | 6 Zealots | 250 |

Table 2: SMAX scenarios used in our experiments.

**Observations**   Each agent observes, for entities within sight range (allies and enemies): *relative coordinates*, *health*, *last_movement_x*, *last_movement_y*, *last_targeted*, and *weapon cooldown*. Self features include *health*, *absolute* $(x, y)$, and *cooldown*.

**Action Space**   Actions are {up, down, left, right} (4 moves), stop, and attack for each attackable enemy unit. Invalid actions are masked before the policy's selection, following SMAC/SMAX defaults.

**Reward**   Per episode, the environment reward comprises: (i) *damage shaping* proportional to total damage dealt to enemies (the total sum is 1), and (ii) a *win bonus* of 1. The maximum cumulative environment reward is therefore 2. No additional reward shaping from environment is applied.

**CTDE Access and Centralized Critic**   Under CTDE, the centralized critic consumes the global state $(S_t^e, S_t^x)$ with the same per-entity features as the agents' observations, *excluding* the history features *last_movement_x*, *last_movement_y*, and *last_targeted*, matching the SMAX implementation. Actors are trained with per-agent shaped rewards; the critic regresses returns computed from their average (see Sec. 4).

**Endogenous and Exogenous State Factors**   In SMAX, the decomposition of the state into endogenous and exogenous coordinates admits a natural and symmetric interpretation grounded in the underlying game mechanics. Endogenous variables correspond to those state components whose next-step values are directly affected by the agents' joint action. For the controlled ally team, executing a move action changes an ally unit's $(x, y)$ position, and executing an attack action modifies the ally's weapon-cooldown timer and reduces an enemy unit's healths. Consequently, the endogenous state in SMAX consists of: (i) ally positions $(x^{\text{ally}}, y^{\text{ally}})$, (ii) ally weapon-cooldown values, and (iii) enemy healths. These quantities lie on the causal pathway of the agents' actions and therefore represent the controllable component of the transition model.

By symmetry, the exogenous state comprises the complementary set of variables whose transitions are governed by the scripted opponent and are not influenced within a single step by the ally agents' actions. Enemy positions and weapon-cooldowns evolve solely according to the built-in opponent micro-policy, and ally health change only due to enemy attacks. Thus, (i) enemy positions $(x^{\text{enemy}}, y^{\text{enemy}})$, (ii) enemy weapon-cooldowns, and (iii) ally healths together form the exogenous state. These components reflect environment-driven or opponent-driven dynamics and act as external sources of non-stationarity from the learners' perspective. This symmetric partition aligns precisely with the ED-POMDP interpretation, in which an agent's action influences endogenous coordinates immediately while affecting exogenous factors only indirectly through delayed interaction with the environment.

---

[1] https://github.com/FLAIROx/JaxMARL

## B.2 SMAX STRATEGY CONFIGURATIONS

A Strategy $m$ is a Cartesian tuple with the following axes:

$$enemy\_shoots \in \{T, F\}, \quad attack\_mode \in \{closest, weakest, random\},$$

$$persistency \in \{T, F\}, \quad move\_mode \in \{center, ally, random\},$$

$$target\_mode \in \{closest, weakest, random\}, \quad distance \in \{4, 6, 8\}.$$

If *enemy_shoots* = F, *attack_mode* and *persistency* are inactive; the total number of strategy settings is $3 \times 2 \times 3 \times 3 = 54$ when *enemy_shoots* = T, plus $3 \times 3 = 9$ when *enemy_shoots* = F, totaling **63** strategies. We further cross strategies with *distance* $\in \{4, 6, 8\}$ (spawn proximity), yielding $63 \times 3 = \mathbf{189}$ configurations.

**Spawn Proximity (distance)** We parameterize the initial ally-enemy distance via $\rho \in \{0.4, 0.6, 0.8\}$ (abbreviated as 4/6/8). Given map width $W = 32$, the nominal distance is $d^\star(\rho) = W \cdot \rho + 2$ (the $+2$ ensures a minimum separation after per-unit noise). All ally (enemy) units are then placed by adding i.i.d. noise of $\pm 1$ around their team center; thus the realized distance is close to $d^\star(\rho)$ up to this jitter. We fix the configuration for the entire episode and sample a fresh configuration at the start of each episode.

**Coding** We encode a strategy as a 6-character token: `[S][A][P][M][T][D]`, with S $\in$ $\{T, F\}$ for *enemy_shoots*; A $\in \{C, W, R\}$ for *attack_mode*; P $\in \{T, F\}$ for *persistency*; M $\in \{C, A, R\}$ for *move_mode*; T $\in \{C, W, R\}$ for *target_mode*; D $\in \{4, 6, 8\}$ for *distance*. For example, `TCTCC6` denotes *enemy_shoots*=T, *attack_mode*=closest, *persistency*=T, *move_mode*=center, *target_mode*=center, *distance*=6.

## B.3 TRAIN-TEST SPLITS AND DISTANCE METRIC

We define a simple Hamming distance over the six axes and curate training splits that avoid near strategies. For **small** regime, no distance variation is used. The **medium** regime uses a wider set, while the **large** regime extends medium by adding 20/5 additional train/test strategies respectively to test generalization over wide coverage. The specific list of strategy for each scale is as follows:

**Small (3 train / 3 test)** Train={`TCTCC6, TWTCW6, TRFAR6`}; Test={`TCTCR6, TWFAC6, TRTCW6`}.

**Medium (10 train / 5 test)** Train={`TCTCR6, TCFAR8, TCFRW4, TWTRC6, TWTRR4, TWFAC8, TRTCC8, TRTCW6, TRFAW4, FXXAR6`}; Test={`TCTCC4, TCTAW6, TWTCW8, TWTAC4, TRFCC4`}.

**Large (30 train / 10 test)** Train={`TCTCW4, TCTCR6, TCTAC6, TCTRC6, TCTRR4, TCFCW6, TCFAC8, TCFAR8, TCFRW4, TWTCR4, TWTRC6, TWTRR4, TWTRR6, TWFCC8, TWFAC8, TWFAR6, TWFRC8, TRTCC8, TRTCW6, TRTAW8, TRTRW6, TRFCW8, TRFAW4, TRFRC8, TRFRR6, FXXAW4, FXXCR4, FXXCC8, FXXRW8, FXXAR6`}; Test={`TCTCC4, TCTAW4, TCTAW6, TWTCW4, TWTCW8, TWTAC4, TWTAC6, TWTRW6, TRTCR4, TRFCC4`}.

For completeness, we also report standard SMAX results without opponent variation in Appendix F, using the built-in heuristic opponent configuration corresponding to `TCTCR6`. All other settings follow the same observation, action masking, and reward specifications as above.

## B.4 EVALUATION PROTOCOL AND COMPUTATIONAL COST

At evaluation, a configuration is sampled uniformly from the designated test set and held fixed for the full episode; we aggregate 256 episodes per evaluation. All results are averaged over 8 random seeds. We report average win-rate performance for SMAX.

The experiments were conducted on various systems equipped with NVIDIA RTX 3090 (24GB) or NVIDIA RTX 4090 (24GB) GPUs. End-to-end wall-clock: The main results were conducted from

the system with Intel(R) Xeon(R) Gold 6326 CPU and 8 NVIDIA RTX 3090 GPUs, $\leq$4 hours for LEICA and $\sim$26h for other baselines.

Under identical training conditions, LEICA incurs additional computational cost compared to MAPPO due to its extra components. Detailed measurements show that the overhead is modest in small-agent settings (e.g., 13% in 3s_vs_6z) and increases with the number of agents, reaching up to 97% in the largest scenarios. In practice, this corresponds to an increase from roughly 5 to 10 minutes per $10^7$ environment steps. Given the substantial performance gains, this level of overhead is not a limiting drawback.

## C APPENDIX: IMPLEMENTATION DETAILS

All experiments use JAX/Flax with JaxMARL-style runners and vectorized environments, collecting fixed-length rollouts from multiple parallel environments and training on minibatches on GPU (Rutherford et al., 2024). We apply parameter sharing for homogeneous agents and action masking at the policy head; centralized training with decentralized execution is used throughout. For MAPPO (Yu et al., 2022), QMIX (Rashid et al., 2018), and SHAQ (Wang et al., 2022), we follow the JaxMARL implementation. For LAIES (Liu et al., 2023), as there was no code implemented in a version suitable for SMAX, we re-implement on top of QMIX by adding the original intrinsic influence term from the authors' code and paper,[2] keeping the QMIX mixing and target update structure (Liu et al., 2023). We implement M3DDPG (Li et al., 2019) by following EPyMARL (Papoudakis et al., 2020) and the official M3DDPG codebase [3], reusing only the buffer and sampling structure of QMIX implementation while re-implementing the algorithmic components. For M3PPO, we modify MAPPO to learn a state-action value instead of a state value, and apply the same max-min update scheme used in M3DDPG.

## D APPENDIX: HYPERPARAMETERS

Hyperparameters were tuned with Bayesian optimization over a fixed budget (Snoek et al., 2012), and search spaces are detailed below.

### D.1 DEFAULT PARAMETERS

| Hyperparameter | Value |
|---|---|
| Discount factor $\gamma$ | 0.99 |
| GAE parameter $\lambda$ | 0.95 |
| Value loss coefficient | 0.5 |
| Minibatch size | 1 |
| Critic Network | [128, 128] |
| Actor Network | [128, 128] |
| Recurrent Network | GRU |
| Recurrent Hidden | 128 |
| Activation | ReLU |
| **LEICA specific** | |
| $\alpha$ | exponentially decaying $\alpha$ |
| Reward weight $\lambda$ | 0.2 |
| EMA ratio $\beta$ | 0.2 |
| Softmax $\tau$ | 0.5 |

Table 3: MAPPO / M3PPO / LEICA default hyperparameters

---

[2] https://github.com/liuboyin/LAIES
[3] https://github.com/dadadidodi/m3ddpg

| Hyperparameter | Value |
|---|---|
| Discount factor $\gamma$ | 0.99 |
| GAE parameter $\lambda$ | 0.95 |
| Buffer batch size | 64 |
| Network size | [128, 128] |
| Mixer embedding size | 32 |
| Mixer hypernet size | 64 |
| Recurrent Network | GRU |
| Recurrent Hidden | 128 |
| Activation | ReLU |
| Maximum gradient norm | 10 |
| Minimum exploration coef $\epsilon$ | 0.05 |
| Epochs | 4 |

Table 4: QMIX / LAIES / SHAQ / M3DDPG default hyperparameters

## D.2 HYPERPARAMETER SEARCH SPACES

| Hyperparameter | Value |
|---|---|
| Entropy Coefficient | {0, 0.01} |
| Learning Rate | {2e-4, 5e-4, 1e-3, 2e-3} |
| Maximum gradient norm | {1, 5, 10} |
| PPO clipping $\epsilon$ | {0.05, 0.1, 0.2} |
| PPO epochs | {4, 8, 16} |
| Rollout length | {64, 128} |
| **M3PPO specific** | |
| Adversarial rate | {1e-1, 1e-2, 1e-3} |
| **LEICA specific** | |
| latent $z, y$ size | {8, 16, 32} |
| Sampling number $K$ | {4, 8, 16} |
| Reconstruction weight $\lambda_{recon}$ | {1e2, 1e3, 1e4} |
| KL regularization $\beta_z, \beta_y$ | {0.5, 1} |

Table 5: MAPPO / M3PPO / LEICA hyperparameter search spaces

| Hyperparameter | Value |
|---|---|
| Learning Rate | {2e-4, 5e-4, 1e-3, 2e-3} |
| Rollout length | {8, 16, 32} |
| Replay buffer size | {1000, 2000, 4000} |
| Target update interval | {100,200,400,800} |
| **SHAQ specific** | |
| Sample size | {1, 5, 10} |
| **M3DDPG specific** | |
| Adversarial rate | {1e-1, 1e-2, 1e-3} |

Table 6: QMIX / LAIES / SHAQ / M3DDPG hyperparameter search spaces

### D.3 GUIDELINES FOR HYPERPARAMETER SELECTION IN LEICA

**Influence-weight parameters** $(\tau, \beta)$ The softmax temperature $\tau$ controls how selectively the weights focus on influential endogenous coordinates. We find empirically that values in the range 0.5-1.0 work well without introducing sensitivity issues. Since performance is largely unaffected once the endogenous coordinates become distinguishable, this parameter can be tuned by gradually decreasing its value while monitoring whether the coordinate structure remains stable. The EMA coefficient $\beta$ (typically 0.1-0.5) smooths weight updates and improves stability. Its effective behavior is correlated with the update interval (or batch size): when updates occur frequently with small batches, a lower $\beta$ is preferable, whereas larger batches or slower update intervals allow for higher $\beta$. Although this parameter generally has limited impact on overall performance, it can influence early-stage training stability.

**Intrinsic reward weight** $\lambda$ The intrinsic reward should guide adaptation without overwhelming the task reward. Moderate values such as $\lambda \in [0.1, 0.3]$ work reliably in SMAX, but this coefficient is generally a sensitive parameter and should be tuned per environment. In practice, the appropriate scale depends on both the magnitude of state changes and the range of extrinsic rewards. For SMAX, where per-step rewards lie roughly in $[0, 1]$ and endogenous state changes are modest, $\lambda = 0.1$ proved effective. However, environments with larger reward scales or larger state variations may require substantially different values.

## E LLM USAGE

We used LLMs only for polishing text. All algorithmic design, mathematics and experiments were written and verified by the authors.

## F ADDITIONAL EXPERIMENTAL RESULTS

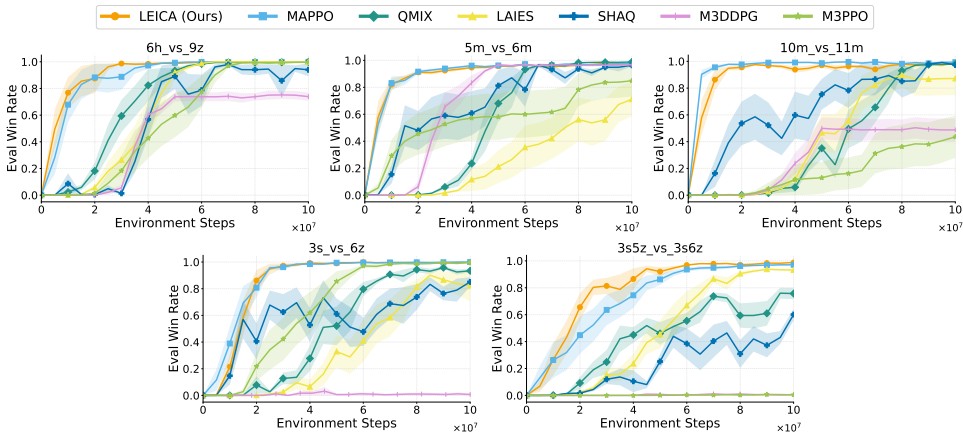

Figure 6: **Single-task SMAX.** Learning curves on standard SMAX **without** exogenous variation. LEICA matches or exceeds baselines, indicating that the proposed method does not degrade conventional performance.

**Single-Task Verification on Standard SMAX** On the original single-strategy SMAX benchmarks, LEICA maintains competitive or superior performance relative to all baselines, as evidenced in Figure 6. This confirms that the proposed design is not specialized solely for distribution shifts but also remains effective in fixed-opponent settings.

**Additional Results per All Task-Regime Pairs**   We provide all task results with 8 random seeds below. Fig. 7, 8, 9 represents the results for Small, Medium, Large scale respectively.

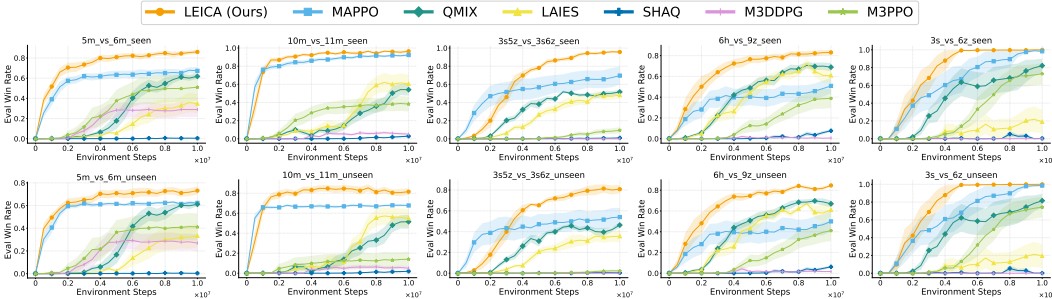

Figure 7: Average win rate for all task-regime pairs in the small-scale benchmark.

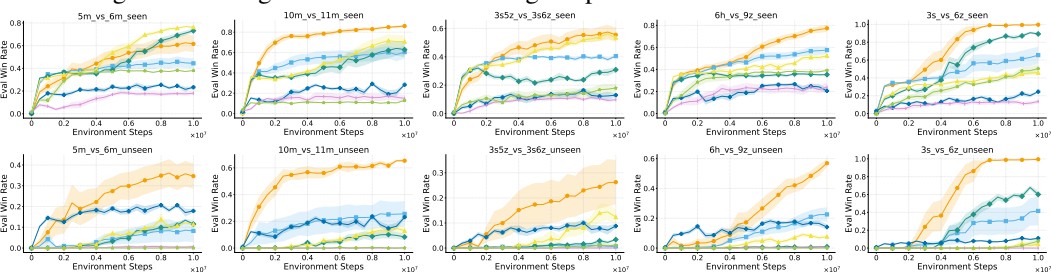

Figure 8: Average win rate for all task-regime pairs in the medium-scale benchmark.

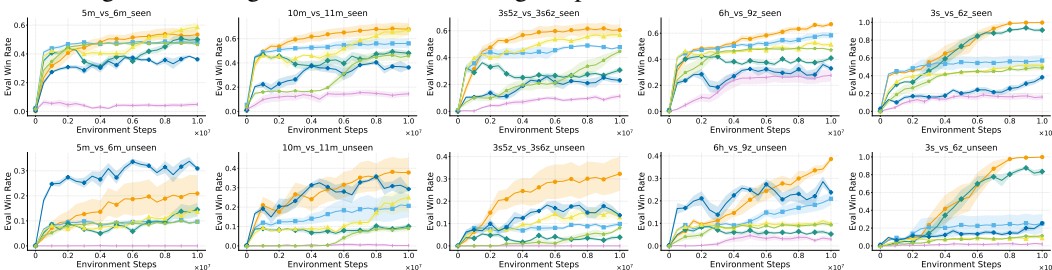

Figure 9: Average win rate for all task-regime pairs in the large-scale benchmark.

