# OpenReview forum: "Addressing Exogenous Variability in Cooperative Multi-Agent Reinforcement Learning"
_ICLR.cc/2026/Conference — Submitted to ICLR 2026_

### Official Review · Reviewer_i8ij · 2025-10-18

**Soundness:** 2
**Presentation:** 3
**Contribution:** 2
**Rating:** 2
**Confidence:** 4

**Summary:**

The paper addresses generalization in multi-agent reinforcement learning (MARL) and proposes Exogenous Decentralized POMDPs (ED-POMDP) to model the separate endogenous influences of the agents' actions and the exogenous influences by the environment (transition). Based on a modeled causal chain, Learning Exogenous Influence for Coordination and Adaptation (LEICA) is proposed to shape rewards with respect to the separated influences. An endogenous and exogenous state predictor based on variational autoencoders are trained to estimate causal influences via a Jacobi matrix of the predictors. These influence estimates are transformed into counterfactual intrinsic rewards and added to the extrinsic reward for standard MARL algorithms like MAPPO to train on. LEICA is evaluated on a Jax-based variant of StartCraft Multi-Agent Challenge (SMAX) and shown to generalize to randomized variations of the original micromanagement tasks, where initial states are varied to different degrees.

**Strengths:**

The paper focuses on an important and interesting topic. Generalization in MARL, especially regarding distribution shifts, are open challenges that need to be addressed.

The paper is well-written and mostly easy to follow.

**Weaknesses:**

**Novelty**

While the addressed problem is well-motivated, the approach is merely a mixture of existing techniques that are already known/established:
1. Multi-task reinforcement learning [1,2,3]
2. Initial state variation [4,5,6]
3. Reward shaping with counterfactuals/causal inference [7,8,9]

While 3. seems to be a theoretically sound approach to solving the addressed problem, 1. and 2. are known to merely shift the actual problem without sufficiently solving it in a causal manner [10], as the learning algorithms depend on i.i.d. samples (either data point-wise or problem-wise).

To improve the paper, a discussion and experimental comparison to these works is required to justify the proposed approach.

The paper also ignores prior work on robust MARL, which addresses adversarial behavior of separate agents [11,12,13].

**Soundness**

The paper focuses on a causal chain that determines the effect of the endogenous context on the next exogenous context. However, according to Fig. 1, both contexts form a colliding node in the causal graph, which could interfere with the general concept. Since the paper does not provide any theoretical analysis, e.g., regarding identifiability, I cannot confirm the validity of the proposed approach from a causal perspective.

The paper proposes to consider "baseline actions" for the counterfactual rewards, which resemble the "default actions" for the difference rewards or aristrocat utilities introduced in [7].

**Quality**

While the paper is generally well-written and presented, the proposed approach introduces a list of hyperparameters, such as $\alpha$ (trade-off exogenous influence and value-sensitivity), $\tau$ for sharpness of the weighting vector, and $\beta$ for decaying the weighting vector. $\lambda$ for the influence-based reward, indicating a tuning-intensive approach.

Of these hyperparameters, only $\alpha$ is ablated. For revision, I recommend:
1. A more thorough ablation study on all of these hyperparameters, e.g., for the appendix.
2. An intuition, how to set these parameters, e.g, regarding domains different from SMAX
3. Ideally, finding an (adaptive) approach that can set these hyperparameters automatically.

**Significance**

The experiments are conducted on some well-known SMAC maps and compared to standard MARL baselines, such as MAPPO and QMIX. The only variation tested in the paper is the initial state, which has been investigated for SMAC in [4,5,6], where MAPPO and QMIX have already been demonstrated to perform poorly.

To improve the significance of the work, especially regarding generalization, I suggest the following:
- Vary the rewards and other functions, as defined in Section 3.1
- Compare with the methods introduced in [4,6]
- Compare with adversarial (test) methods introduced in [11,12,13]
- Test other domains beyond StarCraft II, such as multi-agent MuJoCo and Google Research Football

**Literature**

[1] Omidshafiei et al., "Deep Decentralized Multi-task Multi-Agent Reinforcement Learning under Partial Observability", ICML-17

[2] Li et al., "Multi-task Reinforcement Learning in Partially Observable Stochastic Environments", JMLR-09

[3] Hessel et al., "Multi-Task Deep Reinforcement Learning with PopArt", AAAI-19

[4] Lyu et al., "On Centralized Critics in Multi-Agent Reinforcement Learning", JAIR-23

[5] Ellis et al., "SMACv2: An Improved Benchmark for Cooperative Multi-Agent Reinforcement Learning", NeurIPS-23 Benchmarks

[6] Phan et al., "Attention-Based Recurrence for Multi-Agent Reinforcement Learning under Stochastic Partial Observability", ICML-23

[7] Wolpert et al., "Optimal Payoff Functions for Members of Collectives", Advances in Complex Systems 2001

[8] Li et al., "Automatic Reward Shaping from Confounded Offline Data", ICML-25

[9] Jaques et al., "Social Influence as Intrinsic Motivation for Multi-Agent Deep Reinforcement Learning", ICML-19

[10] Schölkopf, "Causality for Machine Learning", 2019

[11] Li et al., "Robust Multi-Agent Reinforcement Learning via Minimax Deep Deterministic Policy Gradient", AAAI-19

[12] Phan et al., "Resilient Multi-Agent Reinforcement Learning with Adversarial Value Decomposition", AAAI-21

[13] Li et al., "Byzantine Robust Cooperative Multi-Agent Reinforcement Learning as a Bayesian Game", ICLR-24

**Questions:**

1. According to the problem statement in Section 3.1: What is the difference to multi-task learning, given that the reward function can vary as well?
2. In Definition 1: What does the letter $I$ in the definition of $S^{e}_{t}$ stand for? Is it the same $I$ (mutual information) as in Equations 1 and 2?
3. Why is the test considered to be adversarial? The experimental setting description implies that the initial states are merely randomized (without any value-minimizing intentions).

---

> ### Author Response · Authors · 2025-11-21
>
> We thank the reviewer for the thoughtful and constructive comments. In the revised paper, the modified text is highlighted in red. We address the reviewer’s comments below.
>
> ## W1. Novelty / Q1. Multi-task RL
>
> > We agree that our initial draft did not sufficiently articulate how our problem setting differs from multi-task RL, initial-state variation, and classical reward-shaping methods. We clarify that:
> >
> > **(1) Not multi-task RL:**
> > Multi-task RL aims to obtain a common policy capable of performing all tasks when multiple MDPs are given. Unlike Multi-task RL, which does not consider OOD tasks, we theoretically and experimentally account for unseen opponent strategies. In this regard, our work differs from Multi-task RL and instead shares similarities with Meta learning, as mentioned in the related work section.
> >
> > **(2) Not initial state variation:**
> > Our variation is not simple randomization of starting conditions. SMAX exposes 189 exogenous configurations via opponent strategy × distance variations, which produce different transition dynamics, not different initial conditions. We clarified this distinction in Sec. 3.1 and Sec. 6.
> >
> > **(3) Regarding the reward shaping works:**
> > Our research has very different setting and objective from the studies the reviewer referenced. Furthermore, designing intrinsic rewards is a common research topic across various fields, and such approaches are also included among the baseline algorithms we compared.
> >
> > **(4) Regarding the robust MARL methods:**
> > Following the suggestion, we implemented both M3DDPG and a PPO-based minimax variant (M3PPO) to test our setting. As shown in the revised main paper and appendix, these robust MARL methods did not perform well under the large family of scripted exogenous opponent regimes. Moreover, we refer the reviewer to our **Common Response** and the updated **Sec. 6.2** of the revised paper, where we explain why minimax-style robustness is structurally insufficient in ED-POMDP.
> >
> >We hope this clarifies the distinct contribution of the ED-POMDP formulation and the LEICA algorithm, beyond existing paradigms.
>
> ## W2. Causal soundness
>
> > **(1) Figure clarification:**
> > We corrected the caption of causal diagram to avoid the misunderstanding and clarified the intended two-step structure (endogenous → future exogenous).
> >
> > **(2) Identifiability / causal soundness:**
> > We agree that full identifiability guarantees are outside our scope. Our goal is not causal discovery but local causal sensitivity estimation using differentiable predictors, similar in spirit to influence functions in supervised learning. Most importantly, our method relies only on observable local sensitivities, and does not rely on structural identifiability. This fits within the well-established “local causal influence” perspective widely used in MARL and representation-learning settings (e.g., influence-based rewards, conditional sensitivity models).
> >
> > **(3) Relation to aristocrat/difference utilities:**
> > Yes, our baseline action resembles the “default action” concept. However, this is just a design choice suited to our causal-influence objective, rather than a conceptual connection to COMA. We expanded Appendix C to make this distinction explicit, and we appreciate the reviewer’s comment for helping us clarify this point.
>
> ## W3. Quality - Hyperparameter concerns
> >Thanks for raising this point. In response, we have added detailed guidelines on how to set hyperparameters on Appendix D.3.
>
>
> ## W4. Significance
>
> > We thank the reviewer for these suggestions.
> > - Vary the rewards and other functions: We have already conducted experiments under these settings. Changes in the opponent strategy affect the transition as a whole, presenting a challenge that is difficult to resolve using existing MARL methods.
> > -   Different from [4-6]: Our variation is not initial state randomization but transition-level opponent-script variation. We clarify this clearly in Sec. 6.
> > -   Comparison to robust baselines [11-13]: We implemented M3DDPG and a PPO-based minimax baseline within the same SMAX framework; results are now included.
> > -   Other domains: We agree this is important future work. SMAX was chosen due to its large exogenous variation and controllable scale. Due to time and computing resource constraints, we were unable to extend the experiments to other environments, but we will actively strive to do so during the discussion period.
>
> ## Q2. Definition 1 - What does “I” denote?
>
> > No, here **I** denotes the identity matrix, not mutual information. We revised the notation to prevent this confusion.
>
> ## Q3. Why call the test “adversarial”?
>
> > Our “adversarial test” refers to adversarial opponent scripts, not adversarial attacks in the robust RL sense. This is a misunderstanding caused by terminology, thus, all terminology has now been revised to prevent confusion.

---

### Official Review · Reviewer_xiHW · 2025-10-29

**Soundness:** 3
**Presentation:** 3
**Contribution:** 2
**Rating:** 4
**Confidence:** 3

**Summary:**

Most MARL approaches are trained against a single fixed adversarial strategy, leaving teams vulnerable to adversarial strategy shifts at test time. In this work, the authors recast cooperative MARL from a new perspective into an Exogenous Dec-POMDP. It separates agent-controllable endogenous and environment-driven exogenous dynamics in order to learn policies. It consists of VAEs to learn endogenous and exogenous dynamics and influence-based reward design. Through experiments in a modified SMAX, the authors demonstrate the effectiveness of their proposed approach.

**Strengths:**

1. This paper views MARL cooperation from a somewhat new perspective.
2. The experimental results demonstrate the effectiveness of the LEICA in a modified SMAX (more challenging with changing opponent strategies)
3. The paper is well-written, especially the introduction.

**Weaknesses:**

1. It seems that decoupling the controllable and non-controllable idea is similar to the idea of DRIMA, which considers environmental risk and cooperation risk.
2. It seems that dividing the state/observation into endogenous and exogenous parts depends on the environment. In SMAX/SMAC, the state/observation can be decomposed in such a way due to the data structure design. However, it is unclear whether it is suitable for other environments.
3. The reward design is similar to the design of COMA. Moreover, formulas (5)-(8) do not provide much insight regarding endogenous and exogenous parts.

REFERENCE

[1] Disentangling Sources of Risk for Distributional Multi-Agent Reinforcement Learning, ICML 22.

**Questions:**

1. line 66, why "a new scalable SMAX"? Do you show that your new SMAX is more scalable through experiments?
2. Figure 1, please describe the blue line and the black line in detail.
3. line 165-166, is the training process divided into multiple stages? The first stage learns the VAEs?
4. line 173, “decomposed as S_t = (...”. Does this rely on the data structure of environments?
5. What are the MARL insights regarding (5)-(8)?
6. line 278-280, There are many MARL value-based approaches published after 2020.

---

> ### Author Response · Authors · 2025-11-21
>
> We thank the reviewer for the thoughtful and constructive comments. In the revised paper, the modified text is highlighted in red. We address the reviewer’s comments below.
>
> ## W1. Similarity to DRIMA
>
> > We thank the reviewer for pointing out DRIMA. Although both works involve “decomposition,” the problem setting and objectives are fundamentally different:
> >- DRIMA considers settings with perturbations in team agent actions and  performs risk decomposition: it separates environmental risk vs. cooperation risk at the action-value distribution level.
> >- LEICA considers exogenous variable setting, where the opponent or environmental dynamics change, and performs transition-level causal factorization: it separates endogenous vs. exogenous state transitions to handle non-stationary exogenous dynamics.
> >
> > Thus, DRIMA decomposes risk in value space, whereas ED-POMDP decomposes the generative process of the state. These approaches are complementary rather than directly comparable.
>
> ## W2 / Q4. Partition depends on the environment (endogenous/exogenous split)
>
> > We fully agree that automatically learning the endogenous/exogenous decomposition would greatly improve generality. We explicitly strengthened the discussion(Sec 7.) about learning the partition as an important future direction.
> >
> > We would like to emphasize that our main contributions lie in the ED-POMDP formulation and the LEICA algorithm. A full clarification is included in the revised paper and summarized in Common Response 1. We thank the reviewer for this insightful question, which helped us improve the clarity of the paper.
>
> ## W3. Reward design similar to COMA
>
>
> >We thank the reviewer for raising this point. As clarified in Appendix C, the resemblance to COMA is only superficial. COMA uses a centralized counterfactual baseline to assign credit by decomposing the joint Q-value into per-agent contributions. In contrast, LEICA does not decompose value at all: each agent computes its own intrinsic reward based on its local endogenous change, but all agents share the same team-level influence weight derived from exogenous and value sensitivity.
>
> > Thus, COMA performs top-down credit assignment from a centralized critic, whereas LEICA performs bottom-up credit shaping where each agent evaluates its own endogenous effect under a shared causal direction. This is a design choice tailored to our goal of capturing endogenous–exogenous influence, rather than a conceptual link to COMA’s advantage-based decomposition.
>
> ## Q1. Line 66 - What does “a new scalable SMAX” mean?
>
> > “Scalable” refers to the environmental scalability: we introduce a large set of 189 adversary variations, enabling broad generalization tests.
>
> ## Q2. Figure 1 - meaning of blue vs. black lines
>
> > In the revision, we made the explanation explicit: the blue arrows highlights the delayed causal influence of endogenous variables on exogenous transitions at $t\rightarrow t+2$, which is what LEICA captures via Jacobian-based sensitivity. We updated the caption accordingly for clarity.
>
> ## Q3. Lines 165–166 - Are there multiple training stages?
>
> > No multi-stage training is used.  LEICA trains the VAEs (endogenous/exogenous predictors) and the policy jointly and in parallel within each update cycle. There is no pretraining phase; this design keeps training scalable and avoids distribution mismatch.
>
> ## Q5. MARL insight from Equations (5)-(8)
>
> >We thank the reviewer for the question. As clarified in the revision, Equations (5)-(8) are not intended to introduce a new MARL insight themselves, but rather to define the shared endogenous influence weights used later in Eqs. (9)-(11).
> >
> >The key idea is that all agents use the same team-level weighting vector $w$, computed from exogenous sensitivity and value sensitivity, rather than per-agent weights. This provides training stability and ensures that intrinsic rewards are aligned across agents, but the equations themselves are mainly a weight-construction step, not a multi-agent mechanism.
> >
> >The actual multi-agent reasoning enters in the subsequent intrinsic reward formulation (Eqs. (9)-(11)), where the shared weight interacts with each agent’s endogenous change.
>
> ## Q6. more value-based MARL works
> > We thank the reviewer for this suggestion.
> > We updated the Related Work section to include several value-based MARL approaches and positioned our method relative to them. Please recommend additional studies if the reviewer believes that are important or relevant.

---

### Official Review · Reviewer_YKj2 · 2025-10-31

**Soundness:** 3
**Presentation:** 3
**Contribution:** 3
**Rating:** 6
**Confidence:** 4

**Summary:**

This paper targets cooperative multi-agent reinforcement learning under exogenous variability—e.g., unseen opponent strategies at test time. The authors propose an Exogenous Dec-POMDP (ED-POMDP) that factorises the global state into an endogenous component (controllable by the team) and an exogenous component (driven by the environment/opponent). Under this formalism they derive LEICA, a CTDE algorithm that (i) learns separate variational predictors for endogenous and exogenous transitions, and (ii) shapes a counterfactual, influence-weighted intrinsic reward which encourages actions that simultaneously improve team return and shift future exogenous states in a favourable direction. Extensive experiments on a new SMAX benchmark with 63 opponent strategies show large gains over MAPPO, QMIX, LAIES and SHAQ in both training and zero-shot generalisation regimes.

**Strengths:**

Novel conceptual framing: ED-POMDP explicitly disentangles controllable vs. uncontrollable dynamics, giving a principled way to reason about robustness to non-stationary, non-learning opponents.
Practical algorithm: LEICA retains the scalability of CTDE actor-critic methods; the additional VAE predictors and Jacobian-based reward are cheap to compute and easy to plug into MAPPO.
Strong empirical results: Across 15 train/test splits (Small / Medium / Large) LEICA consistently outperforms strong baselines, often doubling the win-rate on unseen strategies while maintaining highest training performance.

**Weaknesses:**

Manual state partition: The endogenous/exogenous split is hand-crafted using domain knowledge of SMAX; the method could fail if the partition is misspecified or unavailable in new domains.
Single-step influence: The reward uses only the one-step Jacobian ∂sˆx_{t+2}/∂s^e_{t+1}; long-horizon influence or multi-step planning is not considered.

**Questions:**

1. Could the partition be learned from data using, e.g., conditional independence tests, sparsity priors, or causal discovery?
2. Why restrict influence to one-step? Would a multi-step rollout (even short) improve the credit assignment, especially in sparse-reward tasks?
3. What is the relation between the world model in model-based RL and the proposed exogenous dynamic model?

---

> ### Author Response · Authors · 2025-11-21
>
> We thank the reviewer for the thoughtful and constructive comments. In the revised paper, the modified text is highlighted in red. We address the reviewer’s comments below.
>
>
> ## W1 / Q1. Can the endogenous/exogenous partition be learned from data?
>
> > We agree that the endogenous/exogenous split is provided in our SMAX experiments. This choice is simply a practical instantiation of the ED-POMDP framework in a domain where the endo/exogenous transition factors are cleanly separated. Our contribution is not partition learning itself but the ED-POMDP formulation and the LEICA algorithm that operate when a factorization is given.
> >
> > As noted in the paper, learning the decomposition is one of the most important future directions, and we clarified this further in the revision. We view the ED-POMDP formulation itself as a framework suggesting this decomposition, and we hope the reviewer will recognize that our contribution opens the door to principled mechanisms for discovering such structure in more complex environments.  A full clarification is included in the revised paper and summarized in **Common Response 1**. We thank the reviewer for this insightful question, which helped us improve the clarity of the paper.
>
>
> ## W2 / Q2. Single-step influence
>
> > This is an insightful observation. Our choice of a two-step Jacobian was primarily a practical design: it is computationally efficient, stable to estimate, and aligns naturally with the local decision-time exogeneity assumption.
> >
> > While multi-step influence modeling is indeed appealing, it introduces challenges such as compounding model error and higher-order gradients, especially in sparse-reward multi-agent domains. We agree this is a promising extension, and we highlight it as future work in the paper.
>
> ## Q3. Relation to world models in model-based RL
>
> > In model-based RL, the world model aims to reconstruct the entire transition dynamics for planning or value estimation. In LEICA, the exogenous predictor is used only to estimate local influence, i.e., how endogenous coordinates can steer the exogenous process at decision time. We do not generate samples using the learned model for training purposes, as is common in world model studies.
> >
> > LEICA’s exogenous predictor resembles a component of a world model, but the purpose is fundamentally different. However, due to similarities in implementation in the exogenous/endogenous inference of Sec. 4.1, it has been included in the related work to aid in understanding the component.

---

### Official Review · Reviewer_1dpp · 2025-11-05

**Soundness:** 3
**Presentation:** 3
**Contribution:** 3
**Rating:** 4
**Confidence:** 3

**Summary:**

This paper introduces a framework for cooperative MARL under exogenous variability. The authors reformulate MARL as an Exogenous Dec-POMDP that explicitly separates team-controllable and environment related state components. They propose a CTDE-based algorithm, called LEICA, that combines two variational inference for exogenous/endogenous factors with an influence-weighted intrinsic reward that encourages both coordination and adaptability. Experiments on the SMAX benchmark demonstrate gains over standard MARL baselines.

**Strengths:**

- Authors is proposing an interesting approach to separate the learning of agent controlled states and environment controlled states to improve the robustness of MARL
- The paper is well written and easy to follow.

**Weaknesses:**

- No comparison to robust MARL baselines: my main concern with this paper is that it omits comparison to established minimax or distributionally-robust algorithms (e.g.,M3DDPG, “Empirical Study on Robustness and Resilience in Cooperative Multi-Agent Reinforcement Learning”). This limits the strength of the robustness claim.
- Hand-crafted state partition: The endogenous/exogenous split is assumed given; learning this decomposition automatically would improve generality.
- Computational complexity: Two VAEs per agent and Jacobian-based influence estimation introduce significant training overhead compared to MAPPO/QMIX.

**Questions:**

- how do you apply this type of method to environments where it is hard to separate the exogenous and indigenous states (e.g. image-based inputs)
- Can the intrinsic reward be generalized to other environment other than SMAX, such as the ones that has on opponents. It makes sense for SMAX as we are trying to reduce the opponent’s health, so influence is desired. What happens to the other environment (e.g.cooperative navigation)?

---

> ### Author Response · Authors · 2025-11-21
>
> We thank the reviewer for the thoughtful and constructive comments. In the revised paper, the modified text is highlighted in red. We address the reviewer’s comments below.
>
> ## W1. Comparison to robust MARL baselines
>
> >We thank the reviewer for emphasizing the importance of comparing against established robust MARL approaches. Following the suggestion, we implemented both M3DDPG and a PPO-based minimax variant (M3PPO) to test our setting. As shown in the revised main paper and appendix, these robust MARL methods did not perform well under the large family of scripted exogenous opponent regimes.
> >
> >Rather than repeat the full discussion here, we refer the reviewer to our **Common Response** and the updated **Sec. 6.2** of the revised paper, where we explain why minimax-style robustness is structurally insufficient in ED-POMDP.
> >
> >We appreciate the reviewer’s comment. This comparison was indeed crucial, and our additional experiments and textual clarifications were added specifically in response to your feedback.
>
>
> ## W2. Hand-crafted endogenous/exogenous split
>
> > We fully agree that automatically learning the endogenous/exogenous decomposition would greatly improve generality. We explicitly strengthened the discussion(Sec 7.) about learning the partition as an important future direction.
> >
> > We would like to emphasize that our main contributions lie in the ED-POMDP formulation and the LEICA algorithm. A full clarification is included in the revised paper and summarized in Common Response 1. We thank the reviewer for this insightful question, which helped us improve the clarity of the paper.
>
> ## W3. Computational complexity
>
> > We appreciate the reviewer’s concern. We further expanded Appendix B to provide a clearer complexity analysis. As acknowledged, LEICA adds several training-time components compared to MAPPO. However:
> >
> > -   this cost remains comparable to or lower than established multi-agent baselines such as QMIX or MADDPG,
> >
> > -   all additional computation is limited to training only, and inference-time cost is similar to MAPPO since Jacobians are not used during execution.
> >
> >
> > We believe this overhead is acceptable given the significant generalization gains and remains practical for large-scale MARL experiments.
>
> ## Q1. Application to environments where the split is unclear (e.g., images)
>
> > Thank you for the question. LEICA does not require exogenous/endogenous factors to be observable at the raw input level. For high-dimensional observations such as images, one could first obtain a latent representation (e.g., via a CNN encoder) and define or learn the partition within that latent space.
> >
> > As mentioned in the response to W2, we expanded the discussion in the revision to clarify that LEICA naturally extends to latent-level decompositions, and that discovering this decomposition automatically is an important direction for future work.
>
> ## Q2. Generality of the intrinsic reward beyond SMAX
>
> > Our intrinsic reward applies whenever exogenous variability can be meaningfully defined. SMAX provides a particularly clear example, as opponent configurations form a well-structured exogenous process. As seen in the new experimental results we added in Section 6.3, changes in ally unit positions have a greater impact on the next exogenous transition than enemy healths. This implies that our intrinsic reward aids adaptation to exogenous variability, making it applicable in any environment where exogenous variability can be defined, even scenarios without opponents.
> >
> > In other environments such as cooperative navigation, exogenous variation might arise from environmental dynamics. However, applying LEICA there would require a more sophisticated and possibly learned endogenous/exogenous decomposition, as noted in W2 and Q1.

---

### Author Response · Authors · 2025-11-21
**Common Response to Reviewers**

We sincerely thank all reviewers for their thoughtful and constructive feedback. Key concerns repeatedly raised across multiple reviews included, in particular:
(1) the principled definition and generality of the endogenous-exogenous state partition, and
(2) the relationship of our approach to robust MARL methods.
We have substantially revised and strengthened the paper with new explanations, additional experiments, and refined terminology. The revised paper is now available at openreview as PDF with new or modified portion in red color.

Below we summarize the key clarifications and improvements.

## 1. Endogenous–Exogenous State Partition
>Many reviewers questioned whether the endogenous/exogenous split is hand-crafted or environment-specific. We fully agree that the original explanation was too brief. We significantly expanded and clarified this part.
>- ED-POMDP does not require raw-state partitioning. The decomposition can be defined or learned via latent-space masking. We now state explicitly that learning the partition is a major and intended future direction.
>- SMAX provides a uniquely transparent ally/enemy separation, which we used solely to validate the ED-POMDP formulation. We now provide a clear, mechanistic justification in Appendix B.1 of the revised paper.
>- We emphasize that our contribution lies in ED-POMDP formulation and LEICA, not in the particular SMAX split. The SMAX partition is simply a transparent validation case where controllable vs. uncontrollable transitions are objectively defined.
>
>We have added a detailed explanation of the above discussion to the paper, and we hope this will help resolve confusion regarding the manual structure and clarify the generality of the proposed formalization.

## 2. Comparison to Robust MARL Baselines
>We fully agree that comparison with robust MARL methods is crucial. Following the reviewers' suggestions, we implemented M3DDPG and a PPO-based minimax algorithm designed analogously to M3DDPG.
> ### Why robust MARL is fundamentally insufficient in this setting
>While these baselines are valuable, their failure in this experiment is unsurprising for structural reasons. Robust MARL methodologies assume adversarial action perturbations (adversarial co-learners, policy perturbations). In cooperative CTDE environments like SMAX, opponents' actions are neither controlled nor perturbed, and adversarial attacks are absent by definition. Our setting deals with exogenous MDP perturbations, not behavioral-level adversarial optimization. The environment (opponent scripts, map scaling, distance fluctuations) induces state transition-level volatility, which robust MARL algorithms are not designed to model.
>
>This structural mismatch prevents robust MARL methods from effectively handling exogenous variability, and our empirical results confirm this theoretical expectation:
LEICA consistently outperforms robust MARL baselines for both observed and unobserved exogenous strategies.

## 3. Additional Experiment: Influence-Weighted Intrinsic Reward Visualization
>To improve interpretability and address reviewers’ requests for deeper intuition behind our intrinsic reward design, we added a new visualization to the revised paper (Section 6.3 and Figure 3). This experiment analyzes how the influence weights $w_j$ evolve throughout training in the 3s\_vs\_6z scenario.
>
>The result verifies that LEICA’s intrinsic reward operates exactly as intended. Early in training, influence concentrates on ally positional coordinates, when the value function is unstable and the agent must first learn how its controllable dynamics affect exogenous transitions. Later in training, influence shifts toward enemy-health dimensions, consistent with the exponentially decaying intrinsic coefficient and the growing reliability of the extrinsic value function.
>
>This demonstrates that LEICA first uses intrinsic shaping to learn the structure of exogenous variability, and then naturally transitions to value-driven exploitation.
## 4. Additional Revisions of the Paper
>We marked major revised part of the paper with red texts to easily find out.
>- Figure 6, 7, 8, 9 and Table 1 are updated with M3DDPG/M3PPO results.
>- The draft used the terms "adversarial" and "adversary" to refer to external opponent scenario variations, which differs from how these terms denote adversarial attacks in the robust RL literature. To avoid ambiguity, we replace these terms with "exogenous" or "opponent variation". This prevents confusion with “adversarial attacks” used in prior research.
> - Line 115: $I$ is the identity matrix, not mutual information.
> - Line 150: The explanation of blue arrow in Fig 1.
> - Line 527: Limitation part.
> - Appendix B.4: More detail information of computational cost.
> - Appendix D.3: Guidelines to select hyperparameters of LEICA.

We thank the reviewers once again for their thoughtful comments. If any additional concerns arise, we would be more than happy to address them.

---

### Meta-Review · Area_Chair_vD5Z · 2026-01-06

**Summary:**

This paper focuses on the challenge of generalization in cooperative multi-agent reinforcement learning (MARL) when faced with exogenous variability such as shifting opponent strategies at test time.
The authors introduce the Exogenous Dec-POMDP (ED-POMDP) framework, which factorizes state dynamics into team-controllable (endogenous) and environment-driven (exogenous) components.
Building on this, they propose LEICA, an algorithm that learns these separate dynamics via variational predictors and utilizes an influence-weighted intrinsic reward to encourage coordination and favorable future exogenous states. Evaluation on the SMAX benchmark demonstrates that LEICA achieves significant performance gains in zero-shot generalization to unseen opponent strategies compared to standard MARL baselines like MAPPO and QMIX.

However, in novelty, multiple reviewers questioned the distinction between this work and multi-task RL or existing reward-shaping methods (e.g., COMA, DRIMA). Technically, the endogenous/exogenous split was hand-crafted for SMAX, which limits generality.

**Reviewer Concerns:**

Concerns still outstanding:

Reviewers expressed concern that the endogenous/exogenous split was hand-crafted for SMAX. This limits generality.
The authors clarified that while SMAX provided a transparent validation case, the ED-POMDP framework can extend to latent-space masking for high-dimensional inputs like images. Authors explicitly identified automatically learning the partition as a primary direction for future work.

**Reviewer Scores:**

Reviewer i8ij with originally 2 score (Reject) had significant concerns regarding novelty and causal soundness. The concerns are hard to be solved even if they had been able to participate fully in the discussion.

---

### Decision · Program_Chairs · 2026-01-26

Reject